# NME3 is a gatekeeper for DRP1-dependent mitophagy in hypoxia

Chih-Wei Chen [1], Chi Su[1], Chang-Yu Huang [1], Xuan-Rong Huang[1], Xiaojing Cuili [1], Tung Chao [1], Chun-Hsiang Fan[1], Cheng-Wei Ting[1], Yi-Wei Tsai[2,3], Kai-Chien Yang[2], Ti-Yen Yeh[4], Sung-Tsang Hsieh [4], Yi-Ju Chen [5], Yuxi Feng[6], Tony Hunter [7] & Zee-Fen Chang [1,8] ✉

NME3 is a member of the nucleoside diphosphate kinase (NDPK) family localized on the mitochondrial outer membrane (MOM). Here, we report a role of NME3 in hypoxia-induced mitophagy dependent on its active site phosphohistidine but not the NDPK function. Mice carrying a knock-in mutation in the *Nme3* gene disrupting NME3 active site histidine phosphorylation are vulnerable to ischemia/reperfusion-induced infarction and develop abnormalities in cerebellar function. Our mechanistic analysis reveals that hypoxia-induced phosphatidic acid (PA) on mitochondria is essential for mitophagy and the interaction of DRP1 with NME3. The PA binding function of MOM-localized NME3 is required for hypoxia-induced mitophagy. Further investigation demonstrates that the interaction with active NME3 prevents DRP1 susceptibility to MUL1-mediated ubiquitination, thereby allowing a sufficient amount of active DRP1 to mediate mitophagy. Furthermore, MUL1 overexpression suppresses hypoxia-induced mitophagy, which is reversed by co-expression of ubiquitin-resistant DRP1 mutant or histidine phosphorylatable NME3. Thus, the site-specific interaction with active NME3 provides DRP1 a microenvironment for stabilization to proceed the segregation process in mitophagy.

Selective removal of damaged mitochondria by mitophagy is critical for maintaining the quality of mitochondria after stress-induced injury[1,2]. Failure to remove dysfunctional mitochondria by mitophagy leads to the development of neuronal degeneration and cardiomyopathy diseases[3,4]. The molecular steps that eliminate damaged mitochondria involve DRP1-mediated segregation of damaged subdomains from the mitochondrial network, the recruitment of LC3-labeled phagophores via a variety of autophagic receptors to generate eat-me signals, lysosome fusion, and engulfment[1,2]. Alternatively, DRP1-dependent mitochondrially-derived vesicles (MDVs) might be directly engulfed by lysosomes[5]. While many autophagy receptors that mediate mitophagy have been identified and mechanistically studied, the regulatory pathway that controls DRP1 for stress-induced mitophagy has not been well explored.

NDPK family enzymes utilize phosphorylated histidine as a high-energy intermediate in the catalysis of NTP formation from NDPs[6]. NME1 and 2 are two major NDPK members in the cytoplasm that not only supply NTPs in a cell but also regulate different cellular processes via transferring the high-energy phosphate from the catalytic histidine to other proteins[7-11]. NME4 is another NDPK member located in the

[1]Institute of Molecular Medicine, College of Medicine, National Taiwan University, 10002 Taipei, Taiwan. [2]Institute of Pharmacology, College of Medicine, National Taiwan University, 10002 Taipei, Taiwan. [3]Department of Medical Research, National Taiwan University Hospital, 10002 Taipei, Taiwan. [4]Institute of Anatomy and Cell Biology, College of Medicine, National Taiwan University, 10002 Taipei, Taiwan. [5]Institute of Chemistry, Academia Sinica, 11529 Taipei, Taiwan. [6]Experimental Pharmacology Mannheim, European Center for Angioscience (ECAS), Medical Faculty Mannheim, Heidelberg University, 68167 Mannheim, Germany. [7]Molecular and Cell Biology Laboratory, Salk Institute, La Jolla, CA 92037-1002, USA. [8]Center of Precision Medicine, College of Medicine, National Taiwan University, 10002 Taipei, Taiwan. ✉e-mail: zfchang@ntu.edu.tw

intermembrane space of mitochondria where it regulates the fusion of the mitochondrial inner membrane by charging GTP for activation of the GTPase function of OPA1[12], while NME3 is localized on the mitochondrial outer membrane via its N-terminal region. We have previously shown that homozygous mutation at the initiation codon of *NME3* is associated with neuronal degeneration and neonatal death[13]. Fibroblasts derived from the patient displayed defects in mitochondrial dynamics and starvation-induced mitochondrial elongation, which are reversed by the re-introduction of wild-type or catalytic-dead but not an oligomeric-defective mutant of NME3. Our recent study further showed that the N-terminal region of NME3 is capable of binding to phosphatidic acid (PA) on mitochondrial tips, where NME3 exerts mitochondrial tethering function to stimulate MFN-mediated mitochondrial fusion independent of the catalytic phosphorylated histidine 135[14].

Our previous report, on the other hand, found that the expression of WT but not catalytic-inactive H135Q NME3 mutant rescues the survival of patient cells after glucose starvation, suggesting the importance of its catalytic function in cell viability under energy stress. Here we further investigated the function of NME3 in hypoxia, which is a general stress in physiological tissues. We found that NME3 is critical for hypoxia-induced mitophagy dependent on the catalytic histidine 135 site but not its NDPK function, and demonstrated its physiological significance in a mouse model. Our investigation revealed that hypoxia treatment increased DRP1 interaction with histidine phosphorylatable NME3 in a PA-dependent manner, which is essential for DRP1-dependent mitophagy. Since the amount of active DRP1 was reduced in NME3-deficient cells, our study further demonstrated that MUL1, a ubiquitin E3 ligase, is involved in the reduction of active DRP1 and the impairment of hypoxia-induced mitophagy. By ubiquitination experiments, we showed that MUL1 is capable of ubiquitinating DRP1 at specific sites. MUL1 overexpression is sufficient to suppress hypoxia-induced mitophagy, which is reversed by overexpression of a DRP1 mutant resistant to MUL1-mediated ubiquitination or histidine phosphorylatable NME3. Overall, our data suggest that hypoxia stress causes mitochondrial damage with PA induction on MOM, where histidine phosphorylatable NME3 forms a complex with DRP1. This process prevents MUL1-mediated ubiquitination of DRP1, thus enabling selective mitophagy.

## Results
### NME3 is essential for hypoxia-mitophagy via phosphorylatable histidine

To know whether NME3 plays a role in quality control of mitochondria via mitophagy, we first tested whether NME3 can affect hypoxia-induced mitophagy because hypoxia is physiologically present in tissues. HeLa cells expressing the mt-Keima mitochondrial reporter[15] were transfected with control or NME3 siRNA, and then incubated in normoxia or 0.5% oxygen hypoxia chamber. Confocal microscopy analyses of lysosomal mt-Keima fluorescence revealed that hypoxia-induced mitophagy was clearly seen at 24 h in control but not in NME3 knockdown cells (Fig. 1a). Hypoxia treatment also markedly induced LC3B phagophore puncta staining in control cells but little in NME3 knockdown cells regardless of chloroquine (CQ) treatment to block lysosome function (Supplementary Fig. S1a), suggesting a critical role of NME3 in hypoxia-induced LC3B phagophore puncta formation. Western blot analysis of these cells showed similar levels of NIX, accompanied by HIF-1α and BNIP3 upregulation during hypoxia (Fig. 1a). It has been reported that FUNDC1 is a mitophagy receptor in hypoxia[16]. The levels of FUNDC1 were similar in control and NME3 knockdown cells. Consistent with the previous report, FUNDC1 was down-regulated in these cells after hypoxia to prevent non-selective mitophagy[17]. The viability of NME3 knockdown cells was obviously reduced, probably due to the lack of mitophagy (Fig. 1b). To know whether the impairment of mitophagy in NME3 knockdown cells involves an autophagy defect, cells stably expressing GFP-LC3-RPF-

ΔLC3 fusion protein[18] were used to determine autophagy flux. In response to nutrient starvation by Hank's balanced salt solution (HBSS) incubation, control, and NME3 knockdown cells showed similar profiles of HBSS-induced changes of GFP/RFP (Fig. 1c), indicating that NME3 knockdown on its own does not cause autophagy defect.

To delineate the functional elements of NME3 required for mitophagy, wild-type (WT), H135Q (HQ), and E40/46D (ED) mutants of NME3 resistant to NME3 shRNA were re-expressed in HeLa cells infected by lentivirus expressing NME3 shRNA. We have previously shown that the active site H135Q NME3 mutant is catalytically dead, while E40/46D is defective in hexamerization due to weakening of the interface interaction between two trimers of NME3[13]. Remarkably, both the WT and ED mutant of NME3 were able to restore hypoxia-induced lysosomal mt-Keima fluorescence. In contrast, the H135Q mutant had no rescue effect (Fig. 1d). We also compared hypoxia-induced mitophagy in murine embryonic fibroblasts (MEFs) derived from *Nme3*[+/+] and *Nme3*[−/−] mice[19]. Consistently, lysosomal mt-Keima fluorescence was detected in wild-type but not *Nme3* knockout cells, and re-expression of WT or ED, but not H135Q mutant, of NME3 was able to rescue hypoxia-induced mitophagy in *Nme3* knockout MEFs (Fig. 1e). We further used the fibroblasts derived from a patient carrying a homozygous mutation in the initiation codon of *NME3*[13] to perform the same experiment. The results confirmed that hypoxia-induced mitophagy was impaired in the patient fibroblasts, which was reversed by expression of WT or ED mutant NME3 but not the H135Q mutant (Supplementary Fig. S1b, c). Enzymatic assay of purified recombinant WT, H135Q, and ED NME3 proteins revealed that both H135Q and ED mutants lacked NDPK function (Fig. 1f). NDPK members are known to contain high-energy phosphate intermediate at the catalytic histidine residue[6]. Analysis of histidine phosphorylation of these purified proteins by blotting with a 1-pHis-specific monoclonal antibody[20] showed that both WT and ED mutant proteins retained heat-labile 1-pHis modification, whereas the H135Q mutant protein did not (Fig. 1g). The H135Q NME3 mutant is defective in NDPK function and histidine phosphorylation, while the ED mutant loses NDPK function but retains H135 autophosphorylation. Given that ED NME3 mutant is capable of restoring mitophagy, the NDPK function of NME3 appears to be unnecessary for hypoxia-induced mitophagy. Instead, the phosphorylatable H135, is critical for NME3's role in hypoxia-induced mitophagy.

### The physiological importance of catalytic-active NME3

It is well documented that mitophagy plays an essential role in protecting the heart from ischemia-induced infarction and neuronal degeneration[4]. To evaluate the general and physiological importance of functional NME3, we generated homozygous *H135Q* knock-in mutant mice (Supplementary Fig. S2a, b). *H135Q*[+/+] (WT) and *H135Q*[m/m] (H135Q mutant) mice at the age of 12 weeks were similar in body weight (Supplementary Fig. S2c). Their peripheral blood had normal levels of red blood cell, hemoglobin, white blood cell, and platelets (Supplementary Fig. S2d). The effects of acute myocardial infarction after ischemia and reperfusion (I/R) in these mice were then examined. By ligating the left coronary artery for 20 min followed by reperfusion for 24 h, WT mice exhibited smaller heart infarct areas than H135Q mice (Fig. 2a). Of note, the heart weights of these mice were similar (Supplementary Fig. S2e). Considering the contribution of defective mitophagy in age-predisposed neurodegeneration and Purkinje cell abnormalities in newborn patients carrying mutation in *NME3* initiation codon, we further evaluated cerebellar function in *H135Q*[+/+] and *H135Q*[m/m] mouse by performing the ledge test; no difference between *H135Q*[+/+] and *H135Q*[m/m] mice at age of 2-months was observed. However, at age of 9 months, *H135Q*[m/m] mice clearly have an increased score in cerebellar ataxia, suggesting a protective role of catalytic-active NME3 in age-dependent neuronal degeneration disorder (Fig. 2b, Supplementary Movie 1, 2). In addition, *H135Q*[m/m] mice at age of 9 months exhibited a hunchback posture during walking the

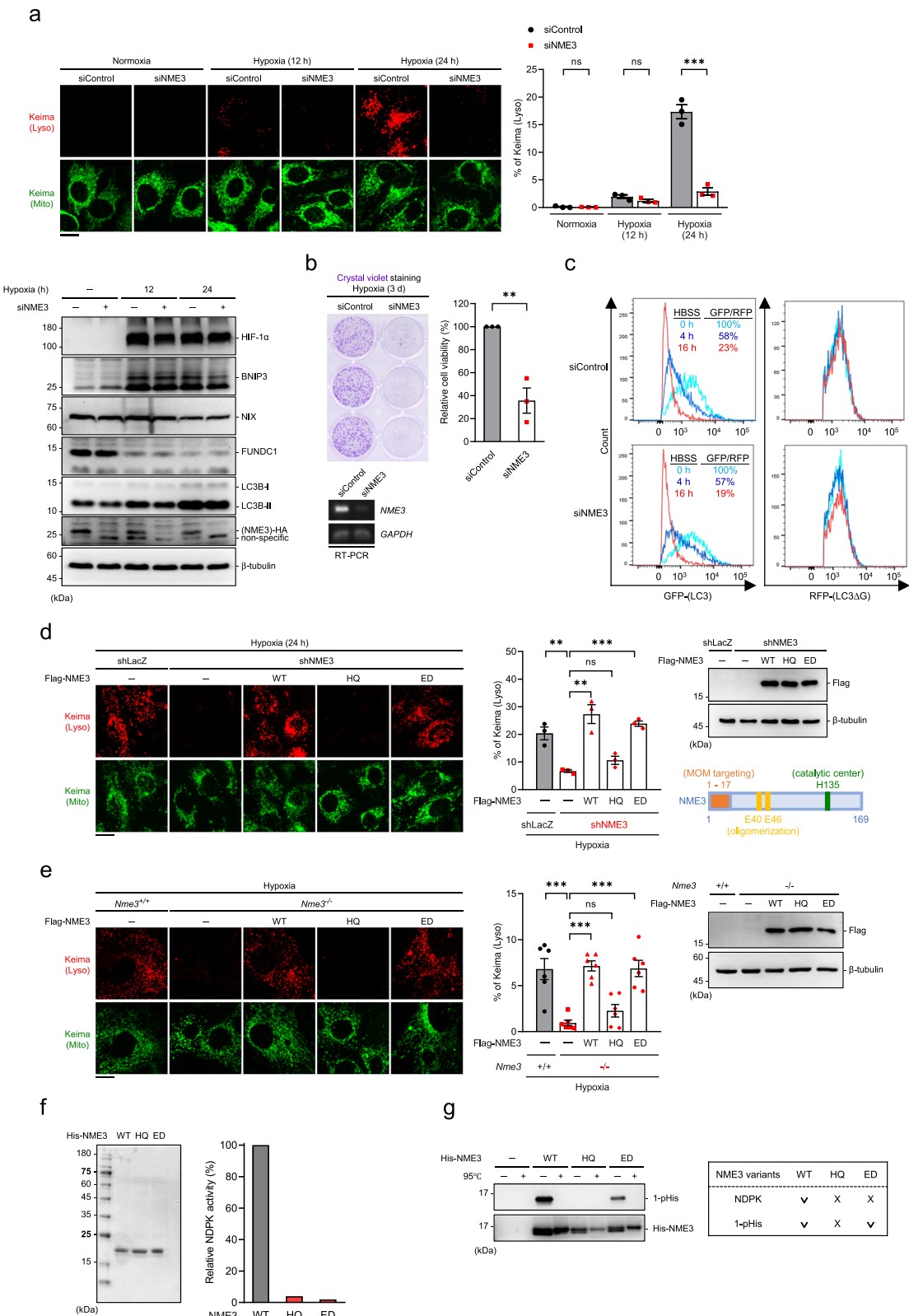

cage's ledge (Fig. 2c). It should be mentioned that NME3 is a short-lived protein in cells and the endogenous level of NME3 is very low. Since a reliable NME3 antibody for detecting endogenous NME3 is not available, we were unable to determine whether tissues from *H135Q^{m/m}* mice still contain NME3 protein. We then performed mass spectrometric analysis of cerebellum isolated from *H135Q^{+/+}* and *H135Q^{m/m}* mice. LC/MS/MS analysis detected NME3 peptides in these tissues. Consistent

with the shorter half-life of H135Q NME3, the amount of detected NME3 peptide from *H135Q^{+/+}* tissue was about 1.6-fold higher than that from *H135Q^{m/m}* tissue (Supplementary Fig. S2f). Therefore, the reduction and the loss of H135 phosphorylation can contribute to both the abnormalities observed in *H135Q^{m/m}* mice.

MEFs isolated from these mice were used to examine viability and mitophagy in response to hypoxia treatment. MEFs from *H135Q^{m/m}*

**Fig. 1 | NME3 is required for hypoxia-induced mitophagy dependent on phosphorylatable H135. a, b** HeLa cells expressing mt-Keima reporters were transfected with control and NME3 siRNA in normoxia and hypoxia. **a** Live cells were visualized by confocal microscopy excited with 488 nm (neutral, green) and 543 nm (lysosomal acidic, red). The percentage of lysosomal (Lyso) Keima of total Keima fluorescence in area was quantitated ($n = 3$ independent experiments). The samples were analyzed by Western blot analysis. **b** After 3 d, cells were fixed and stained with crystal violet ($n = 3$ independent experiments). RT-PCR data of *NME3* and *GAPDH* are shown. **c** Cells stably expressing GFP-LC3-RFP-LC3ΔG transfected with control and NME3 siRNA were starved in Hank's balanced salt solution (HBSS). At the indicated time, cells were harvested for GFP/RFP fluorescence ratio analysis by flow cytometry. **d** The expression of NME3 variants on hypoxia-induced mitophagy. Cells expressing mt-Keima infected with lentivirus of shRNA of *LacZ* and *NME3* were transfected with Flag-NME3 variants resistant to shRNA in hypoxia for

mt-Keima analysis. The representative images, Western blot analysis of the samples, and the quantification of percentage of lysosomal Keima of total Keima fluorescence ($n = 3$ independent experiments) are shown. **e** $Nme3^{+/+}$ and $Nme3^{-/-}$ MEFs stably expressing mt-Keima were transfected with Flag-NME3 variants in hypoxia for 24 h. Representative images, Western blot of the samples, and quantification of lysosomal Keima and total Keima ($n = 6$ independent experiments) are shown. **f, g** Biochemical analysis of NME3 variants. Purified recombinant proteins of His-tagged-NME3 WT, HQ, ED, and control vector were analyzed for **f.** Coomassie blue staining of three purified proteins (*Left*). NDPK enzymatic assay (*Right*). **g** Heat-sensitive analysis of 1-pHis. All samples were incubated with ATP for 30 min and treated with and without 95 °C for 3 min before gel electrophoresis. Samples were analyzed by Western blot of 1-pHis and His-tag., All fluorescence images are shown with scale bar, 20 μm. All quantified data are presented as mean ± SEM. NS means no significance, *$p < 0.05$, **$p < 0.01$, ***$p < 0.001$; two-tailed $t$-test.

mutant mice displayed reduced viability after hypoxia treatment (Fig. 2d). Electron microscopy analysis showed an increased number of membrane-engulfed mitochondria frequently observed in MEFs from $H135Q^{+/+}$ but not $H135Q^{m/m}$ under the hypoxic condition (Fig. 2e). Hypoxia-induced lysosomal mt-Keima fluorescence was seen in $H135Q^{+/+}$ (WT) but not $H135Q^{m/m}$ (HQ) MEFs (Fig. 2f). Both cells had similar levels of HIF-1α induction in WT MEFs compared with that in HQ MEFs. Other NDPK proteins, including Nme1, 2, 4, and mitochondrial outer membrane fission and fusion factors, including Mfn1, Mfn2, Drp1, Fis1, and Mff, in WT and HQ MEFs were also quite similar (Supplementary Fig. S2g). After hypoxia for 2-3 days, the level of VDAC1 was reduced in WT MEFs but remained unchanged in H135Q MEFs (Supplementary Fig. S2h). Moreover, the expression of wild-type NME3 in $H135Q^{m/m}$ MEFs restored hypoxia-induced mitophagy (Fig. 2g, Supplementary Fig. 2i). Altogether, these data confirmed the essential role of phosphorylatable H135 of NME3 in hypoxia-induced mitophagy and its physiological importance.

## Hypoxia-induced PA promotes NME3 interaction with DRP1 for mitophagy

It is known that mitochondrial fission is an important step preceding LC3 binding to trigger piecemeal mitophagy[21,22]. Since NME3 is involved in DRP1-mediated peroxisome fragmentation[23], we speculated that NME3 is required for DRP1 in segregating hypoxia-induced damaged subdomain from mitochondrial network for mitophagy. However, it has been suggested that DRP1 might not be essential for hypoxia-induced mitophagy[24], while another study showed that DRP1 is required[16]. To clarify the role of DRP1 in hypoxia-induced mitophagy, DRP1 was depleted by siRNA in HeLa cells, which impaired mitophagy (Supplementary Fig. S3a). In agreement with FUNDC1 as the mitophagy receptor in hypoxia[16], knockdown of FUNDC1 also abolished mitophagy (Supplementary Fig. S3a). Moreover, we compared mitophagy in $Drp1^{+/+}$ and $Drp1^{-/-}$ MEFs in hypoxia. The results were very clear that mitochondria in $Drp1$ KO MEFs were highly fused and lacked mitophagy signal in hypoxia (Supplementary Fig. S3b), which was reversed by expression of wild-type (WT) but not the K38A mutant of DRP1.

To learn how NME3 is involved in DRP1-mediated mitophagy, we analyzed the interaction of NME3 with DRP1 in response to hypoxia by super-resolution confocal microscopy. Due to the lack of an available NME3 antibody, this study used GFP- and HA knock-ins, separately, to label endogenous NME3 at the C-terminus (Supplementary Fig. S3c, d). The colocalization of endogenous NME3-GFP with DRP1 IF staining were analyzed. The result showed that hypoxia treatment stimulated the interaction of endogenous NME3 and DRP1 (Fig. 3a). Western analysis of endogenous NME3-GFP pulled down with DRP1 confirmed hypoxia-induced complex formation of NME3 with DRP1 (Fig. 3b). By analyzing hypoxia-induced LC3 puncta, the super-resolution microscopy data showed LC3 puncta highly associated with the endogenous DRP1 and NME3-GFP co-localized sites (Supplementary Fig. S3e). These data raised the question why their interaction on mitochondria is increased

by hypoxia. Similar to NME3, DRP1 also has a PA-binding function biochemically[25]. One recent report has indicated that DRP1 mediates formation of mitochondria-derived vesicles (MDVs) for lysosomal degradation in a PA-dependent manner[5], and the proteomic analysis of MDVs also revealed the presence of NME3. Since our recent study has demonstrated that the N-terminal region of NME3 binds to PA on mitochondria[26], we then asked whether hypoxia-induced DRP1/NME3 interaction involves a mitochondrial PA signal. To address this question, we overexpressed Lipin, which converts PA to DAG, to remove the PA signal. In hypoxia, overexpression of WT but not the catalytic-dead mutant of Lipin abolished DRP1 pulldown by endogenous NME3-GFP (Fig. 3c), indicating their interaction is PA-dependent. In the meanwhile, by the expression of PA probes, an EGFP fused with the PA-binding domain of c-Raf1[14], we found that hypoxia treatment markedly increased PA formation on MOM, which was abolished by over-expression of active Lipin (Fig. 3d, e). We then asked the question whether PA is the signal required for hypoxia-induced mitophagy. Data from mt-Keima assays revealed that overexpression of WT but not the catalytic-dead mutant of Lipin diminished the hypoxia-induced mito-phagy signal (Fig. 3f). Since the interaction of the H135Q NME3 mutant with DRP1 was significantly less than that of WT NME3 (Fig. 3g), it is likely that NME3 and DRP1 are recruited to PA sites, where WT- but not H135Q NME3 forms a complex with DRP1. Altogether, these results suggest that a hypoxia-induced PA signal on mitochondria not only promotes the interaction of DRP1 with WT NME3 but also is essential for mitophagy.

We have previously shown that the N-terminal region confers NME3's PA-binding function; however, MOM localization of NME3 via the N-terminal region is not solely dependent on its PA binding[26]. We wanted to distinguish the contribution of PA binding via its N-terminal region or MOM localization of NME3 to DRP1-dependent mitophagy in hypoxia. To this end, the N-terminal region of NME3 was replaced with the mitochondrial transmembrane sequence of TOM20 to enforce its MOM localization without requiring PA binding (Supplementary Fig. S3f). We then expressed WT NME3 or TOM20-NΔ-NME3 in NME3 knockout cells to compare hypoxia-induced mitophagy signal. The results revealed that unlike WT NME3, TOM20-NΔ-NME3 was unable to restore mitophagy signal in NME3 knockout cells (Fig. 3h). Thus, the N-terminal region of NME3 capable of binding to PA sites rather than its MOM localization plays a pivotal role in determining DRP1-dependent mitophagy. Altogether, these results led us to propose that hypoxia treatment generates mitochondria PA sites to dock DRP1 and NME3 and promote their interaction, which is a critical step for DRP1 in selective mitophagy (Fig. 3i).

## A sufficient amount of active DRP1 on mitochondria in hypoxia requires NME3

DRP1 is a GTPase, which mitochondrial membrane scission activity is driven by its GTP-bound form that binds to its fission receptor in an oligomerization manner followed by GTP hydrolysis[27]. DRP1-mediated fission can contribute to mitochondrial proliferation and

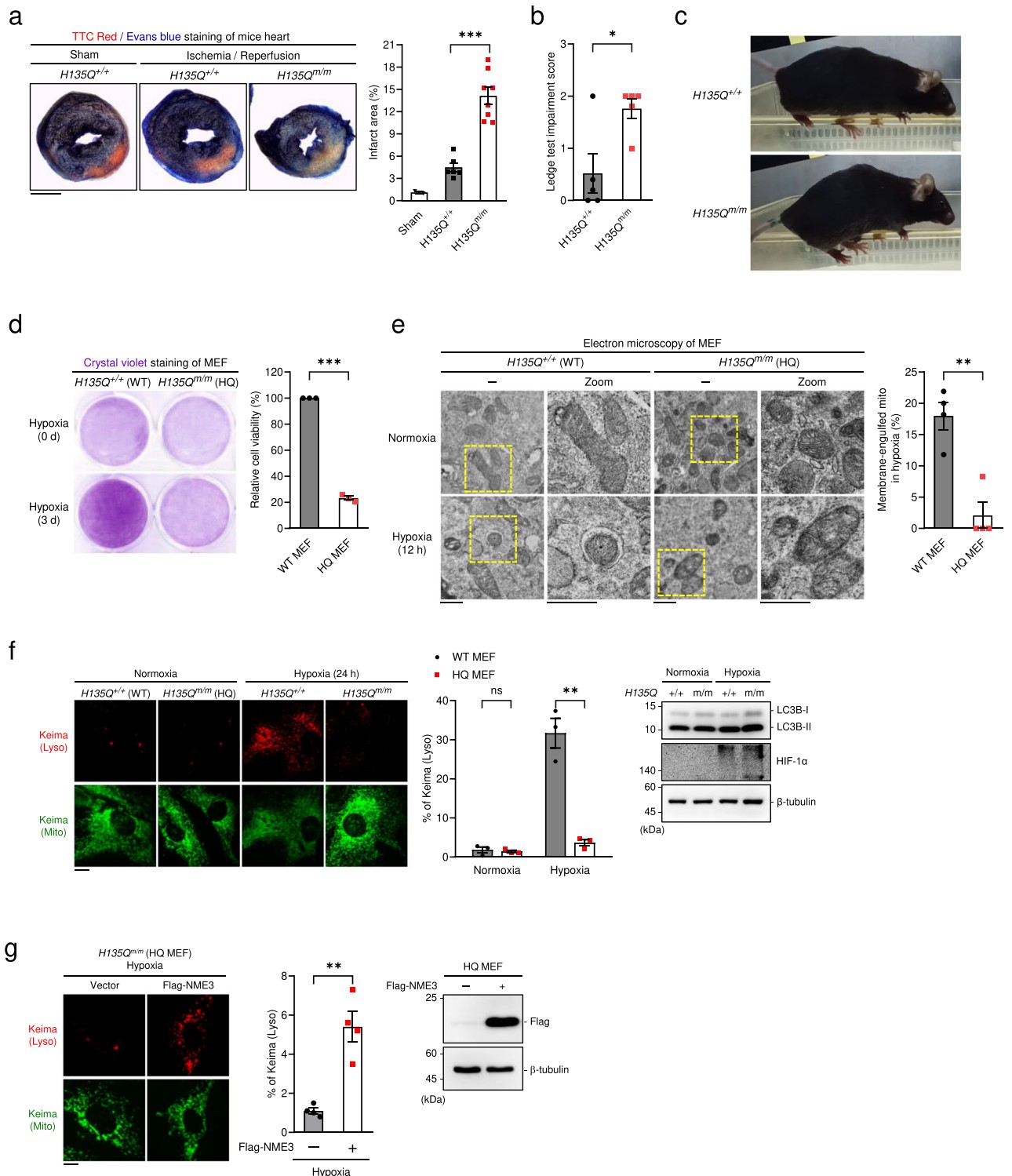

degradation[28]. We then measured the interaction of DRP1 with its fission receptors, FIS1 or MFF, by a proximity ligation assay (PLA). In normoxia, knockdown of endogenous NME3-HA in HeLa cells did not affect the PLA signals of DRP1 interaction with FIS1. In contrast, in hypoxia, the PLA signals of DRP1-FIS were significantly reduced in NME3 knockdown cells (Fig. 4a). The PLA signals of DRP1-MFF were also markedly reduced in NME3 knockdown cells after hypoxia treatment (Fig. 4b). Similar to HeLa cells, the PLA signals of Drp1-Fis1 and Drp1-Mff were higher in WT than those in H135Q mutant MEFs in hypoxia and exhibited no difference in normoxia (Supplementary

Fig. S4a). Clearly, NME3 controls DRP1 function in dividing mitochondria in hypoxia but not in normoxia.

Since NME3 is a NDPK, NME3 association might provide GTP loading for the activation of DRP1. However, the expression of NME3-ED mutant deficient for NDPK function also restored hypoxia-induced mitophagy in NME3-depleted cells (Fig. 1e, f, and Supplementary Fig. S1b, c), suggesting that NDPK activity in GTP loading might not be the key function of NME3 in supporting DRP1 scission in mitophagy. Intriguingly, proteasomal inhibition by MG132 treatment increased the PLA signal of DRP1-FIS1 and DRP1-MFF interaction in NME3 knockdown

**Fig. 2 | The physiological importance of histidine phosphorylatable NME3.**
**a** Wild-type ($H135Q^{+/+}$) and H135Q homozygous knock-in ($H135Q^{m/m}$) mice at the age of 8–12 weeks were subjected to ischemia/reperfusion (I/R) heart injury by occluding the left coronary artery for 20 min followed by reperfusion for 24 h. After sacrifice, the 2,3,5-triphenyltetrazolium chloride (TTC) Red and Evans blue staining was performed for analyzing area of myocardial infarction. Representative sections, scale bar, 2 mm (*Left*). The quantified data of the infarct sizes of hearts from I/R and sham control ($n = 2$ for sham control mice; $n = 6$ for $H135Q^{+/+}$ I/R mice; and $n = 8$ for $H135Q^{m/m}$ I/R mice). **b** Ledge test. Mice walked along the ledge with coordination were assigned a score of 0, while with instances of not using the hind legs properly were scored 1 and fall off scored 2. Each mouse was tested for five rounds for mean score ($n = 5$ mice in each group). **c** The photographs of male mice at the age of 9 months when walking on a cage's ledge. **d**–**f** Wild-type $H135Q^{+/+}$ and

H135Q homozygous knock-in ($H135Q^{m/m}$) MEFs were incubated in hypoxia chamber for **d** viability analysis at the indicated day ($n = 3$), and **e** electron microscopy (EM) analysis post-hypoxia for 12 h, scale bar, 500 nm ($n = 4$). **f** $H135Q^{+/+}$ (WT) and $H135Q^{m/m}$ (HQ) MEFs stably expressing mt-Keima in normoxia and hypoxia 24 h were analyzed. Images of lysosomal Keima and total Keima fluorescence are shown, scale bar, 20 μm, Western blot of the samples, and quantified data ($n = 3$ independent experiments) are shown. **g** $H135Q^{m/m}$ (HQ) MEFs transfected with vector and wild-type Flag-NME3 in hypoxia for 24 h were subjected to mt-Keima analysis. Representative images, scale bar, 20 μm, Western blot of the samples, and quantification data of the percentage of the lysosomal Keima ($n = 4$ independent experiments) are shown. All quantified data are presented as mean ± SEM. NS means no significance, $*p < 0.05$, $**p < 0.01$, $***p < 0.001$; two-tailed $t$-test.

but not control cells, suggesting the involvement of a proteolytic process (Fig. 4c, d). Moreover, overexpression of WT-DRP1 significantly increased hypoxia-induced mitophagy in NME3 knockdown but not control HeLa. In MEFs, DRP1 overexpression also rescued mitophagy signal in H135Q but not WT MEFs (Fig. 4e and Supplementary Fig. S4b). PLA signals of DRP1-FIS1 and DRP1-MFF in NME3-deficient cells were fully restored by DRP1 overexpression (Fig. 4f, g). Of note, mitophagy signal was significantly promoted by DRP1 overexpression in NME3 knockdown HeLa cells, but not fully rescued. Probably, in hypoxia, other factors important for the mitophagy process are also partly regulated by NME3 in HeLa cells. Therefore, DRP1 overexpression was unable to fully rescue mitophagy. Nevertheless, an excess amount of DRP1 is capable of promoting mitophagy in the cells depleted of NME3 but not control cells, suggesting that DRP1 becomes a limiting factor for mitophagy in NME3-defective cells. Since the total amount of DRP1 is not decreased in NME3-defective cells, we then analyzed the amount of active form of DRP1, indicated by pS616, by Western blot analysis in mitochondria-enriched fractions. The result showed that the amount of active DRP1 in the total amount of DRP1 in the mitochondrial fractions was particularly reduced in NME3 knockdown cells (Fig. 4h). Without NME3, DRP1 overexpression compensates for the loss of the active form of DRP1 to promote the dividing step for mitophagy. Since hypoxia-induced NME3 interaction with DRP1 and mitophagy are both PA-dependent, this implies that the NME3/DRP1 complex present in hypoxia-induced PA-enriched microenvironment on MOM is critical for maintaining a sufficient amount of active form of DRP1 for the subsequent segregation.

## DRP1 is susceptible to MUL1 in NME3-defective cells
The above data let us speculate that the association of NME3 might protect active DRP1 from degradation. In particular, we observed that after hypoxic treatment, ubiquitin IF staining associated with mitochondria became very prominent in H135Qm/m MEFs as compared to little signal in WT MEF (Supplementary Fig. S5a), suggesting the possibility that histidine phosphorylatable NME3 might regulate the ubiquitination of proteins on mitochondria under the hypoxia stressed condition. Mitochondria have four major ubiquitin E3 ligases, Parkin[29], MARCH5[30], RNF185[31], and MUL1[32]. Since HeLa cells are deficient for Parkin, the other three E3 ligases were individually depleted to test their involvement in the mitophagy defect in NME3 knockdown cells. The results showed that only MUL1 knockdown restored mitophagy by mt-Keima assay in NME3 knockdown cells (Supplementary Fig. S5b). In MUL1 knockout cells, we confirmed that hypoxia-induced mitophagy was no longer affected by NME3 knockdown (Fig. 5a). Consistently, PLA signals of DRP1-FIS1 and DRP1-MFF interaction in MUL1 KO cells were not reduced by NME3 knockdown in hypoxia (Fig. 5b). Importantly, after NME3 knockdown, the MG132 treatment only increased the DRP1-FIS1 and DRP1-MFF interaction signal in control but not MUL1 KO cells (Fig. 5c). Furthermore, MUL1 knockout increased the level of pS616-DRP1 in mitochondria-enriched fractions in hypoxia-treated NME3 knockdown cells (Fig. 5d). Mul1 knockdown in H135Q mutant MEFs also

restored mitophagy, the PLA signal of Drp1-Fis1 and Drp1-Mff interaction, and pS616-Drp1 level (Supplementary Fig. S5c–f). Super-resolution microscopy analysis further showed that knockdown of Mul1 reduced the ubiquitin signal associated with DRP1 on mitochondria in H135Q MEFs in hypoxia (Fig. 5e). Taken together, these data imply that in NME3-defective cells, DRP1 becomes more susceptible to MUL1 ubiquitination, which restricts the amount of active DRP1 in hypoxia for mitophagy.

## MUL1-mediated DRP1 ubiquitination impairs mitophagy
We further found that NME3 depletion significantly increased the PLA signals of MUL1 and DRP1 interaction after hypoxia treatment (Fig. 6a). In MEFs, the PLA signals of Drp1-Mul1 were very prominent in H135Q mutant but not WT MEFs (Supplementary Fig. S6a). These data confirmed the increase of DRP1 susceptibility to MUL1 in NME3-defective cells. It has been shown that MUL1-mediated Sumoylation stabilizes DRP1 for mitochondrial fragmentation and apoptosis[33,34]. We found that MUL1 overexpression in HeLa cells did increase the PLA signal of DRP1-FIS1 and DRP1-MFF in normoxia. In contrast, in hypoxia, overexpression of MUL1 reduced DRP1-FIS1 and DRP1-MFF interaction (Fig. 6b and Supplementary Fig. S6b). This indicates that MUL1 has an opposite effect on DRP1 in hypoxia condition. In addition, we found that MUL1 overexpression prominently increased the ubiquitin signal associated with mitochondria in hypoxia but not normoxia (Fig. 6c), suggesting that the ubiquitination by MUL1 is stimulated by hypoxia. It is possible that hypoxia-induced ROS might cause a functional shift of MUL1 in the regulation of DRP1.

MUL1 has been shown to be a ubiquitin E3 ligase of a number of proteins including AKT, TP53, MFN2, NF-kB, and ULK1[35]. We then examined whether DRP1 can be ubiquitinated by MUL1. By co-expression of Flag-MUL1 and myc-DRP1 in HEK293T cells, we found that DRP1 was ubiquitinated by MUL1 (Fig. 6d). The database of post-translational modification of proteins showed that DRP1 has multiple ubiquitination sites (Supplementary Fig. S6c). We then generated a series of lysine-to-arginine mutants of DRP1, which were co-expressed with Flag-MUL1 and HA-Ubiquitin (Supplementary Fig. S6d). Among them, only K271/272R myc-DRP1 mutant was resistant to MUL1-mediated ubiquitination (Fig. 6d). In vitro ubiquitination assays also showed that WT-myc-DRP1 was ubiquitinated by MUL1 and much less for K271/272R mutant (Fig. 6e).

To determine whether MUL1-mediated ubiquitination of DRP1 is responsible for impairing mitophagy, we compared the expression of WT and K271/272R mutant myc-DRP1 in MUL1-mediated suppression of mitophagy. Either overexpression of WT or K271/272R myc-DRP1 did not alter hypoxia-induced mitophagy, suggesting that the mutation does not affect the function of DRP1 in mitophagy (Fig. 6f). Next, we compared hypoxia-induced mitophagy in cells co-expressing MUL1 with WT or K271/272R myc-DRP1. Unlike WT-myc-DRP1, co-expression of the K271/272R mutant did not respond to MUL1 overexpression in mitophagy suppression (Fig. 6f), indicating DRP1 ubiquitination at K271/272 by MUL1 overexpression impairs mitophagy. We further analyzed the effect of MUL1 overexpression on DRP1 interaction with MFF in cells

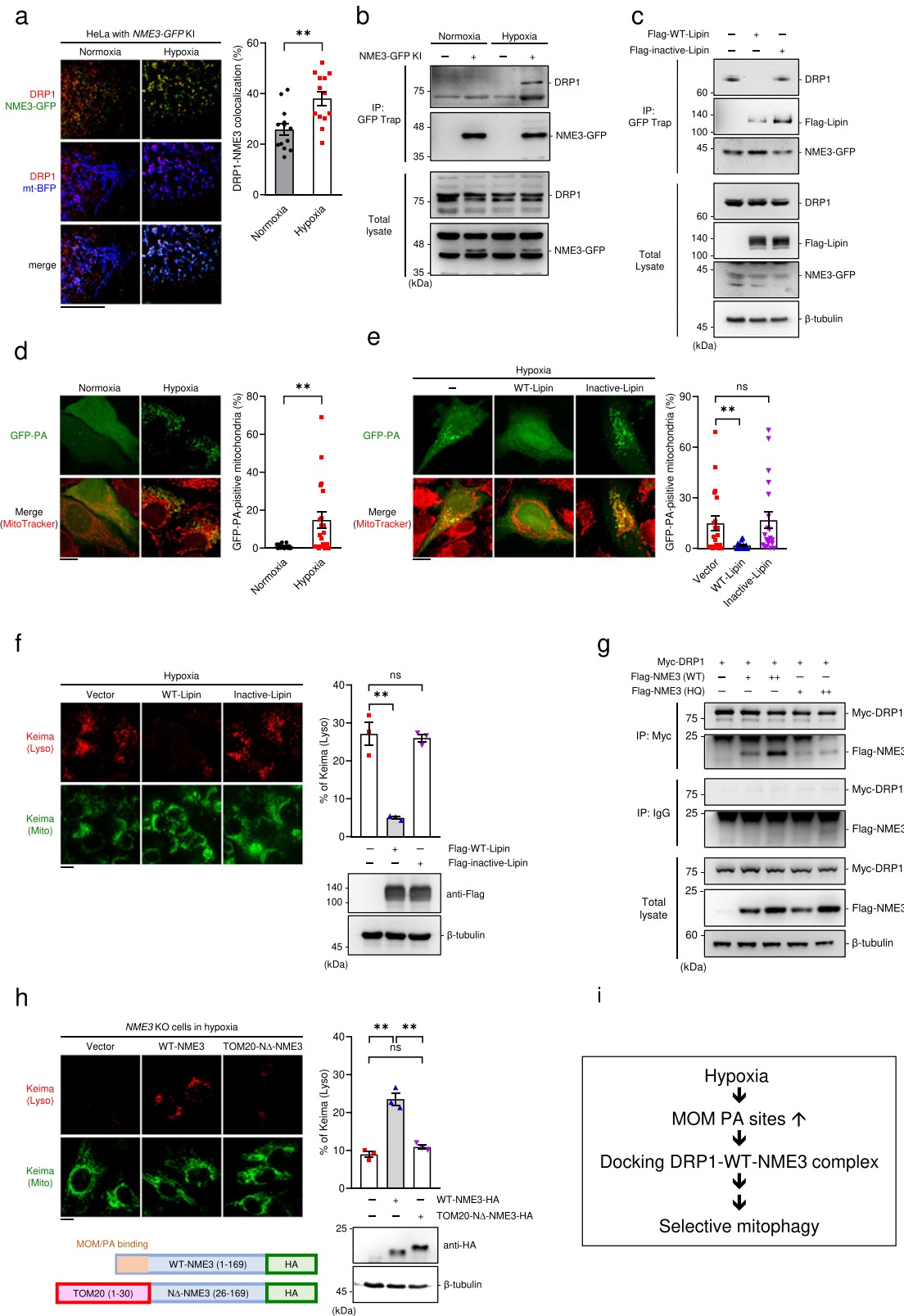

expressing WT and K271/272R mutant of DRP1. Consistent with mitophagy data, overexpression of MUL1 alone markedly reduced DRP1-MFF and DRP1-FIS1 interaction signals (Fig. 6g and Supplementary Fig. S6e). In cells co-expressing K271/272R mutant myc-DRP1, MUL1 overexpression did not reduce the DRP1-MFF PLA signal, in contrast to the prominent reduction in cells overexpressing WT-myc-DRP1. Overall, these results suggest that hypoxia stress-induced mitochondria damage

switches the functional effect of MUL1 on DRP1, leading to MUL1-mediated ubiquitination to suppress DRP1-dependent mitophagy.

## NME3 antagonizes MUL1-mediated DRP1 ubiquitination and mitophagy suppression

Finally, we addressed the question of how NME3 controls MUL1-mediated regulation of DRP1 in hypoxia. In HeLa cells, knockdown of

**Fig. 3 | PA-dependent NME3 and DRP1 interaction and hypoxia-induced mitophagy. a** HeLa-NME3-GFP cells in normoxia and hypoxia overnight were fixed for IF staining and analyzed by AiryScan super-resolution confocal microscopy. Representative images, scale bar, 10 μm (*Left*). Quantitation of the colocalization of DRP1 with NME3-GFP in all DRP1 area (*n* = 13) (*Right*). **b** Cells in normoxia and hypoxia were harvested for NME3-GFP pulldown using GFP-trap. The Western blot of DRP1 in NME3-GFP pulldown. **c** NME3-GFP cells transfected with empty, WT, and catalytic-dead-lipin vectors in normoxia and hypoxia overnight for GFP-trap pulldown and analyzed by Western blot. **d** Cells transfected with GFP-PA plasmid in normoxia and hypoxia overnight were stained with mitoTracker Red and fixed for confocal microscopy analysis. Representative images, scale bar, 20 μm (*Left*). Quantification of the percentage of GFP-PA on mitochondria (*n* = 20) (*Right*). **e** Cells transfected with GFP-PA plasmid in combination with control, WT-, and catalytic-dead lipin vector in hypoxia overnight were fixed for confocal microscopy analysis. Representative images, scale bar, 20 μm (*Left*). Quantification of the percentage of GFP-PA on mitochondria (*n* = 20) (*Right*). **f** Cells expressing mt-Keima transfected with empty, WT-, and catalytic-dead lipin vector in hypoxia were subjected to mt-Keima analysis. Images, scale bar 20 μm, and quantitation of lysosomal Keima and total Keima fluorescence are shown (*n* = 3 independent experiments). Western blot analysis of Flag-lipin expression. **g** HEK293T cells transfected with Myc-DRP1 together with WT, or H135Q (HQ) of Flag-NME3 plasmids were harvested for immunoprecipitation in the presence of 1 mM of ATP and analyzed by Western blot. **h** NME3 knockout HeLa cells expressing mt-Keima transfected with empty vector, WT NME3-HA, and TOM20-NΔ-NME3-HA mutant plasmids were subjected to hypoxia-induced mitophagy analysis. Representative images with scale bar 20 μm, quantitation of lysosomal Keima and total Keima fluorescence (*n* = 3 independent experiments), and Western blot of Flag-NME3 expression. **i** A model in which hypoxia-induced PA promotes NME3 interaction with DRP1 for mitophagy. All quantified data are presented as mean ± SEM. NS means no significance, *p < 0.05, **p < 0.01, ***p < 0.001; two-tailed *t*-test.

NME3 increased MUL1-DRP1 interaction (Fig. 6a). To understand the relationship between NME3 and MUL1, the interaction of NME3 with MUL1 was analyzed. We used NME3-GFP HeLa cells for pulldown analysis by GFP-trap (Fig. 7a, b). The analysis showed endogenous MUL1 pulldown by endogenous NME3-GFP in hypoxia but not in normoxia (Fig. 7a). The pulldown of MUL1 was abolished by the treatment of N-acetylcysteine (NAC) and MitoTEMPO, a mitochondrial antioxidant, during hypoxia (Fig. 7b). These results indicate that hypoxia-induced reactive oxygen species (ROS) promotes NME3 interaction with MUL1.

To investigate if the ubiquitin E3 ligase function of MUL1 is affected by its association with NME3, we further performed an in vitro ubiquitination reaction with GST-MUL1. Since GST-MUL1 is in a dimer form, the E3 ligase activity of MUL1 was evaluated by auto-ubiquitination. GST-MUL1 was pre-incubated with WT or H135Q mutant NME3 in the presence of ATP. Since 1-pHis autophosphorylation of NME3 is an unstable modification, the addition of ATP in the pre-incubation would ensure the 1-pHis modification difference between WT and H135Q mutant of NME3 protein. Glutathione beads pulldown of GST-MUL1 with His-NME3 was used for the in vitro ubiquitination assay. The presence of WT but not H135Q NME3 suppressed auto-ubiquitination of GST-MUL1 on beads (Fig. 7c), suggesting that the interaction with NME3 attenuates the E3 ubiquitin ligase function of MUL1. Since DRP1 preferentially interacts with WT NME3 (Fig. 3g), we then tested whether ubiquitination of DRP1 by MUL1 is affected by the association with WT NME3. The in vitro ubiquitination assay showed that DRP1 ubiquitination by MUL1 was suppressed by the presence of WT NME3 (Fig. 7d).

Relevant to MUL1-mediated ubiquitination stimulated by hypoxia and the negative control of MUL1 by NME3, we found that the mitochondrial ubiquitin signal was increased in NME3 knockdown cells in hypoxia but not in normoxia (Supplementary Fig. S7a). MUL1 knockdown diminished the ubiquitin signal associated with mitochondria in NME3-depleted cells (Supplementary Fig. S7b). We further co-expressed NME3 variants with MUL1 and found that WT- and ED mutant but not H135Q suppressed MUL1-mediated ubiquitin signal associated with mitochondria in hypoxia (Fig. 7e). Since WT and ED mutant of NME3 is histidine phosphorylatable, we further tested the antagonistic effect of histidine phosphorylatable NME3 on MUL1-mediated suppression of hypoxia-induced mitophagy. The mt-Keima assay showed that the mitophagy suppression effect of MUL1 over-expression was reversed by co-expression of WT and ED mutant of NME3 but not H135Q mutant of NME3 (Fig. 8a). Consistently, MUL1-mediated suppression of PLA signal of DRP1-MFF interaction in hypoxia was reversed by co-expression of WT or ED mutant but not H135Q of NME3 (Fig. 8b), suggesting that histidine phosphorylatable NME3 antagonizes MUL1 in disabling DRP1.

In summary, we proposed a model in which DRP1 interacts with histidine phosphorylatable NME3 at hypoxia-induced damage sites enriched with PA lipid on the mitochondrial outer membrane. Active

DRP1 at damage sites is protected from MUL1-mediated ubiquitination by the association with NME3, thereby ensuring a sufficient amount of active DRP1 for segregating the damaged subdomains. Without histidine phosphorylatable NME3, DRP1 binding at the PA site is no longer protected and becomes susceptible to MUL1-mediated ubiquitination. The loss of active DRP1 at damage sites by MUL1-mediated ubiquitination therefore impairs hypoxia-induced selective mitophagy. Thus, NME3 acts as a gatekeeper for DRP1-mediated selective mitophagy in hypoxia. This mechanism might contribute to severe ischemia/reperfusion cardiac injury and early onset of cerebellar ataxia observed in mice carrying H135Q knock-in mutation in *Nme3* (Fig. 8c).

## Discussion

Our findings uncover an unprecedented role of NME3 in the control of DRP1 for hypoxia-induced mitophagy. Beginning with bacteria, NDPK has been shown to have multiple functions, including NTP formation, nuclease, and histidine kinase activity[6,36,37]. In evolution, vertebrates acquired *NME3*, which contains an N-terminal sequence for binding to the mitochondrial outer membrane. Given the high abundance of NME1 and 2 in mammalian cells as compared to NME3, the contribution of NME3 to the NTPs levels is not that important[38]. WT or ED mutant NME3 that retains histidine 135 phosphorylation but not the H135Q NME3 mutant are able to restore hypoxia-induced mitophagy in NME3-deficient cells. Since H135Q and ED mutants of NME3 are both inactive in NDPK function, we conclude that the role of NME3 in hypoxia-induced mitophagy is independent of NDPK function. In agreement with the importance of mitophagy in health, a knock-in H135Q mutation at *Nme3* in mice increases I/R-induced heart infarction and facilitates age-dependent cerebellar ataxia. These findings illuminate the role of histidine phosphorylatable NME3 in mitochondrial quality control.

This study showed that DRP1 is essential for hypoxia-induced mitophagy. Based on the PA-binding activity of NME3, via its N-terminal region, and DRP1 described previously[25,26], we further established the role of a PA signal that mediates the interaction of DRP1 with NME3. Importantly, we found that hypoxia treatment indeed increased PA sites on mitochondria. Since the removal of PA by Lipin overexpression reduced DRP1/NME3 interaction and abolished hypoxia-induced mitophagy, it is apparent that PA-induced site-specific interaction of NME3 and DRP1 participates in a determining step in mitophagy. In support of this notion, we showed that unlike WT NME3, TOM20-NΔ-NME3, which is localized on MOM via the transmembrane domain of TOM20 and lacks a PA-binding domain, was unable to restore hypoxia-induced mitophagy in NME3-deficient cells. Therefore, MOM localization is insufficient for supporting NME3 function for mitophagy; instead, it is the PA-binding function of NME3 critical for DRP1-dependent mitophagy. It has been shown that oligomerization-driven GTPase activity of DRP1 is suppressed by the interaction with PA and saturated acyl chains of phospholipid on liposomes and mitochondria[25]. However, DRP1 has been shown to mediate MDV

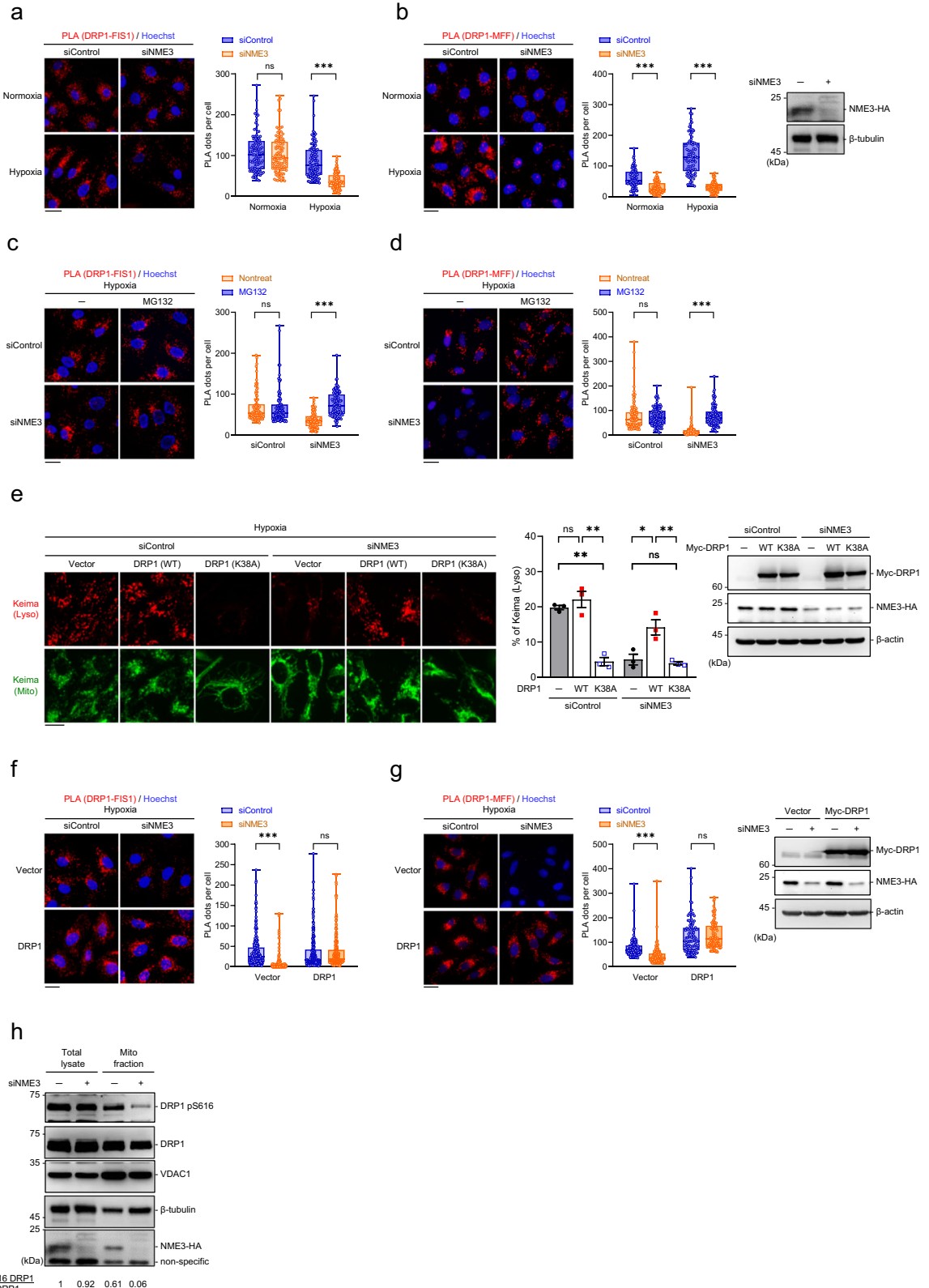

scission dependent on PA, which is a highly enriched lipid in MDV[5]. It is likely that PA microenvironment might dictate the function of DRP1. Presumably, the negative curvature on the mitochondrial surface is involved in membrane scission, which can be very different from the curvature on mitochondrial tips for fusion. Whether the curvature type in PA-rich microenvironment can affect the activity of DRP1 function remains to be investigated.

This study also addressed the question of how PA-dependent interaction with NME3 promotes DRP1 function in mitophagy. The expression of the NME3-ED mutant defective in NDPK function, DRP1 overexpression, and MUL1 depletion all restore DRP1 interaction with fission receptor and hypoxia-induced mitophagy in NME3-defective cells. These data suggest that GTP loading of DRP1 is not necessarily dependent on its association with NME3. Very likely, cytosolic GTP

**Fig. 4 | The reduction of active DRP1 in NME3-defective cells. a–d** DRP1 inter-action with fission receptors. HeLa cells transfected with control and NME3 siRNA in hypoxia for 20 h were analyzed by PLAs of (**a**) DRP1-FIS1 interaction and (**b**) DRP1-MFF interaction. The representative images and quantitation data of PLA dots per cell (total 100 cell for DRP1-FIS1 and 80 cells for DRP1-MFF from 3 independent experiments) are shown. **c, d** Cells with and without siNME3 transfection in hypoxia overnight were treated with MG132 for 4 h before PLA for **c** DRP1-FIS1 interaction and **d** DRP1-MFF interaction. The representative images and quantitation data of PLA dots per cell ($n = 100$) are shown. All box-and-whiskers plots show minima and maxima of all individual points with box ranged from 25th to 75th percentile. **e–g** DRP1 overexpression effects. **e** Cells expressing mt-Keima were transfected with empty, wild-type Myc-DRP1, and Myc-DRP1 (K38A) vectors in hypoxia for 24 h for confocal microscopy analysis. The representative images, Western blot of the

samples, and quantification of the percentage of lysosomal Keima of total Keima fluorescence in area ($n = 3$ independent experiments) are shown. **f, g** Cells were transfected with control and NME3 siRNA with myc-DRP1 vector in hypoxia for 24 h for analyzing the PLA signal of **f** DRP1-FIS1 interaction and **g** DRP1-MFF interaction. The representative images, Western blot of the samples, and quantitation data of PLA dots per cell ($n = 240$ cells for DRP1-FIS1 interaction and $n = 100$ cells for DRP1-MFF interaction) are shown. **h** The level of pS616-DRP1. HeLa cells with NME3-HA knock-in were transfected with siControl and siNME3 in hypoxia for 24 h, followed by mitochondrial isolation using TOM22 microbeads. Total and mitochondria-enriched fractions were analyzed by Western blots using antibodies against pS616 of DRP1, total DRP1, VADC1, β-tubulin, and HA tag. All images are shown with scale bar, 20 μm. All quantified data are presented as mean ± SEM. NS means no sig-nificance, *$p < 0.05$, **$p < 0.01$, ***$p < 0.001$; two-tailed $t$-test.

---

levels are sufficient for DRP1 GTP loading regardless of its association with NME3. Our findings showing the effect of MUL1 depletion in increasing the level of active DRP1 and its interaction with fission receptors in hypoxia point out a negative control of DRP1 by MUL1 in NME3-defective cells. A previous report has shown that MUL1 is a mitophagy suppressor in neuronal cells[39]. Here, we found that increasing expression of MUL1 is sufficient to suppress hypoxia-induced mitophagy and to cause DRP1 ubiquitination at K271 and K272 sites. The suppression of mitophagy or DRP1 binding to fission receptor by MUL1 overexpression is abolished by co-expression of the DRP1 K271/271R mutant. Since K271/272 is located in the GTPase domain, MUL1-mediated DRP1 ubiquitination might cause certain changes in function and degradation as well. It has been shown that MUL1 overexpression gives K48-linked ubiquitin chain of a number of proteins including p53, ULK1, AKT, and Mfn2[35] for proteasomal degradation. Likely, DRP1 located at damage sites without NME3 pro-tection is destabilized by MUL1. In addition to acting as an E3 ubiquitin ligase, MUL1 has been shown to act as a SUMO E3 ligase in DRP1 stabilization to increase mitochondrial fragmentation and apoptotic induction[33,34]. Apparently, the effects of MUL1 on DRP1 are opposite under normoxia, hypoxia, and apoptotic conditions. Since MUL1 is a cysteine-rich E3 ligase, it will be interesting to understand how its function in ubiquitination is stimulated in hypoxia.

The most important question of this study is how NME3 controls the regulation of DRP1 by MUL1. We found that hypoxia-induced ROS promotes the formation of a complex between endogenous NME3 and MUL1. Therefore, one possibility is that NME3 might squelch MUL1 to prevent its accessibility to DRP1 as revealed by the PLA signal (Fig. 6a). Intriguingly, the in vitro ubiquitination assay showed that the ubiquitin E3 activity of MUL1 and MUL1-mediated DRP1 ubiquitination were suppressed by WT NME3. Consistently, in hypoxia, MUL1 over-expression suppressed mitophagy, DRP1 interaction with fission receptors, and the ubiquitin signal associated with mitochondria, all of which were reversed by co-expression of histidine phosphorylatable, active NME3. These data raise the possibility that NME3 acts as a his-tidine kinase to phosphorylate MUL1 and inactivate its E3 ligase activity, and this needs further study.

While the mechanisms of eat-me signals and receptors for mitophagy are well studied, there is an information gap in under-standing the molecular process determining the segregation of the damaged subdomain from the mitochondrial network for mitophagy. Our findings suggest that NME3 is a gatekeeper at PA-enriched sites to warrant the DRP1-mediated segregation process for stress-induced mitophagy.

## Methods
### Antibodies and dyes
The primary and secondary antibodies used for Western blotting and immunofluorescence (IF) staining are shown in the Supplementary Table 1. Dyes used for IF staining include Hoechst 33342 (H1399, Invitrogen, 2 μg/mL) and MitoTracker Red CMXRos (M7512,

ThermoFisher Scientific, 1:1000 dilution in medium and incubated at 37 °C for 15 min before fixation).

### Plasmids
The mitophagy reporter mt-Keima was purchased from Addgene (#131626, pHAGE-mt-mKeima). The bacterial expression of His-tagged NME3 variants by using pET-28m-6xHis-thrombin-NME3 plasmids were previously described[13]. The mammalian expression of pLAS-pPuro-pTK-Flag-NME3-WT, -H135Q, and -E40/46D variants were constructed by insertion PCR products amplified from the tet-on expression plasmids of Flag-NME3s from published reference [13] at *Nhe*I(5′) and *Age*I (3′) of pLAS-pPuro-pTK vector, which contains herpes simplex virus (HSV) thymidine kinase (TK) promoter for the low-level expression of NME3. Human MUL1 was amplified by PCR from HeLa cDNA and inserted into *Sal*I (5′) and *BamH*I (3′) site of pCMV2-FLAG vector or pGEX-4T-1 vector, yielding pCMV2-Flag-MUL1 and pGEX-4T-a-MUL1 plasmids, respectively. All mutations were generated by using a QuikChange II XL Site-Directed Mutagenesis Kit (200522, Agilent). Mutations H319A of MUL1 was generated by designed primers: Forward: 5′- GGT GCA GGA ACA AAC <u>GGC CCC</u> ACA CTC CAG AAA G −3′, and Reverse: 5′- CTT TCT GGA GTG TGG <u>GGC</u> CGT TTG TTC CTG CAC C −3′. pMRX-GFP-LC3-RFP-ΔLC3 (Addgene #84572), pRK5-Flag-wild-type and catalytic-dead lipin 1 plasmids were purchased from Addgene (Addgene# 32005 for WT-lipin, and #32006 for catalytic-dead-lipin). TOM20-NΔ-NME3-HA plasmid was generated by replacing the N-terminal 1-25 amino acid region of WT NME3-HA in pLAS2w.pPuro-pTK vector with the N-terminal 1-30 amino acid of TOM20. RFP-ubiquitin and mitoYFP are kindly provided by Ruey-Hwa Chen (Academia Sinica, Taiwan). The pGW1-Myc-DRP1 and pEGFP-C1-Raf1-PABD plasmids are a generous gift from Chuang-Rung Chang (National Tsing Hua University, Taiwan) and Ya-Wen Liu (National Taiwan University) according to published references [14,40], respec-tively. All the mutants of Myc-DRP1 at lysine sites were generated by site-directed mutagenesis.

### Cloning, expression, and purification of recombinant proteins and NDPK activity assay
pGEX-4T-1-MUL1-WT, -H319A, pET-28m-6xHis-thrombin-NME3-WT, -H135Q, and -E40/46D were separately transformed into *Escherichia coli* BL21(DE3) strain. IPTG (isopropyl β-D-1-thiogalactopyranoside, 0.5 mM) was added to the bacterial culture from single colony for further incu-bation at 30 °C for 3 h. Bacterial pellet was resuspended in the lysis buffer (50 mM Tris-HCl pH 8.0, 50 mM NaCl, 1% NP-40, 1 mM 1,4-dithiothreitol, 1% protease inhibitor, and PMSF). For His-tagged NME3 (His-NME3) protein, 1 mM of imidazole was included in the lysis buffer. Following sonication of pellets, the lysates were centrifuged at 13,800 × $g$ for 15 min at 4 °C. The supernatants were transferred to a tube containing Ni-NTA Agarose (1018244, Qiagen) and Glutathione Sepharose 4B (17-0756-01, GE Healthcare) beads to pulldown His-NME3 and GST-MUL1, respectively. After 1.5 h of gentle rotation at 4 °C, Nickle beads were centrifuged at 400 × $g$ for 2 min with subsequent washing by

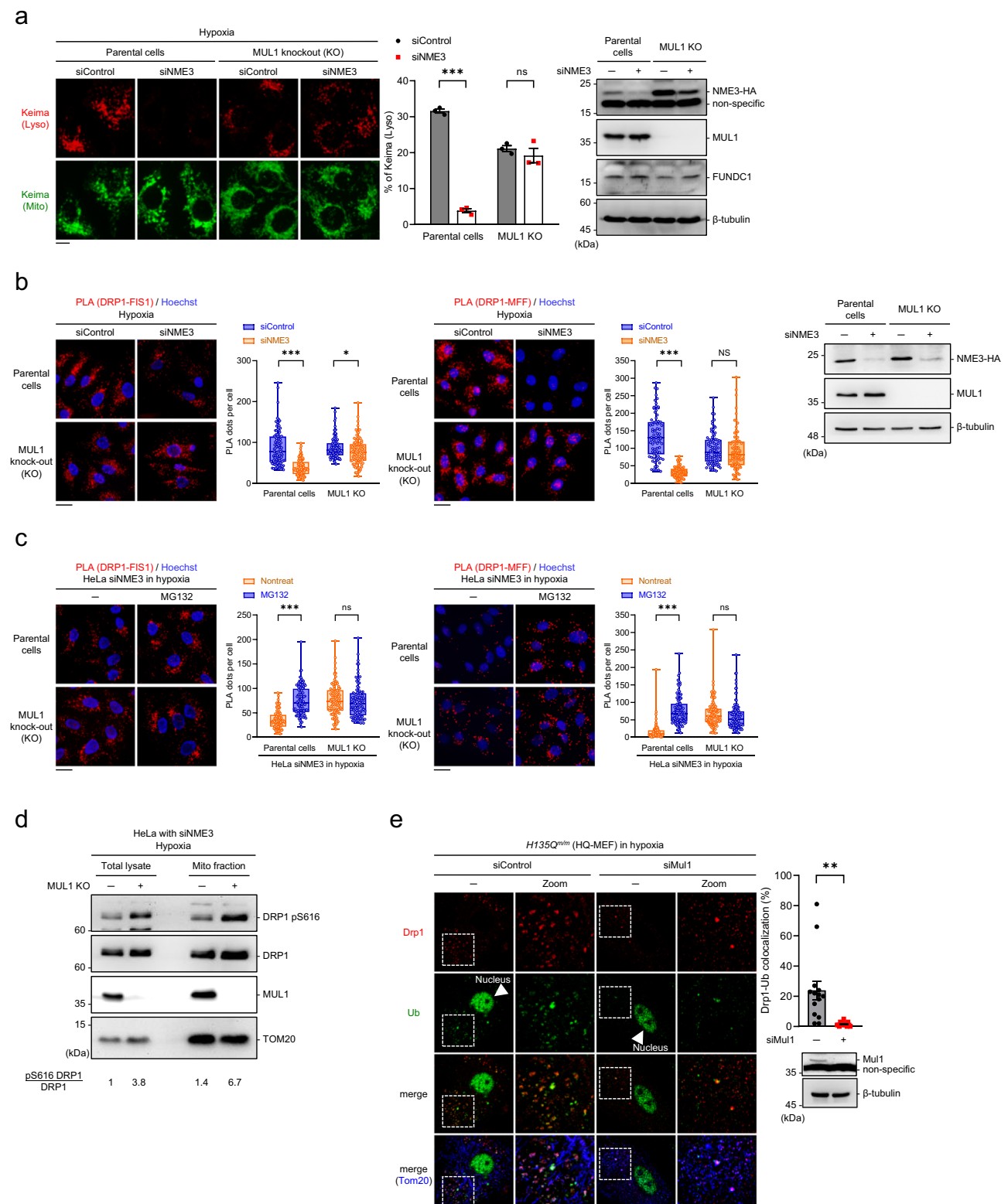

the buffer (50 mM Tris-HCl pH 8.0, 50 mM NaCl, 5 mM imidazole, 1% NP-40, protease inhibitor, and 1 mM PMSF) for five times. The Myc-DRP1 in pET30a plasmid was cloned and transformed to *Escherichia coli* BL21(DE3) strain. His-Myc-DRP1 protein was purified as described above. GST-MUL1 protein was eluted with the buffer (50 mM Tris-HCl pH 8.8, 150 mM NaCl, 5 mM MgCl$_2$, 1 mM 1,4-dithiothreitol, and 10 mM glutathione). His-NME3 protein was eluted by the buffer containing 50 mM Tris-HCl pH 7.5, 50 mM NaCl, and 500 mM imidazole followed by

dialysis in buffer containing 50 mM Tris-HCl pH 7.5, and 50 mM NaCl. NDPK activity assay was as previously described[13].

### Cell culture, transfection, and infection

HeLa and 293T cells were maintained in Dulbecco's modified Eagle's medium (DMEM) supplemented with 10% heat-inactivated fetal bovine serum (HI-FBS), 100 U/mL penicillin, and 10 μg/mL streptomycin. Mouse embryonic fibroblasts (MEFs) were maintained in DMEM with

**Fig. 5 | MUL1 impairs hypoxia-induced mitophagy in NME3-defective cells.**
**a** Parental and MUL1 knockout (KO) HeLa cells expressing mt-Keima were transfected with control and NME3 siRNA for 2 d. After hypoxia for 24 h, the lysosomal and total Keima fluorescence were analyzed. The representative images, Western blot of the samples, and quantification data ($n = 3$ independent experiments) are shown. **b** After control and NME3 siRNA transfection for 2 d and hypoxia for 24 h, HeLa cells were fixed for PLA of DRP1-FIS1 and DRP1-MFF interaction. The representative images, Western blot of the samples, and quantitation data of PLA dots per cell ($n = 100$ cells for DRP1-FIS1 and $n = 80$ cells for DRP1-MFF) are shown. **c** After hypoxia, NME3 knockdown in parental and MUL1 KO cells were treated with MG132 for 4 h before fixation for PLA of DRP1-FIS1 and DRP1-MFF interaction. The representative images and quantitation of PLA foci in each cell ($n = 100$ cells from 3 independent experiments) are shown. All box-and-whiskers plots show minima and maxima of all individual points with box ranged from 25th to 75th percentile.
**d** NME3 knockdown in parental and MUL1 knockout cells were harvested for preparing mitochondria-enrichment fraction. Western blot of pS616-DRP1, total DRP1, MUL1, and TOM20 are shown. **e** Colocalization of Drp1 with ubiquitin (Ub). After hypoxia, $H135Q^{m/m}$ MEFs with and without Mul1 knockdown were fixed for IF staining of Drp1, ubiquitin, and Tom20 for AiryScan super-resolution microscopy analysis. The images of colocalization of Drp1 with ubiquitin and Tom20 and the quantitation data ($n = 14$ cells) are shown. Images are shown with scale bar, 20 μm. All quantified data are presented as mean ± SEM. NS means no significance, $*p < 0.05$, $***p < 0.001$; two-tailed $t$-test.

10% HI-FBS, 100 U/mL penicillin, 10 μg/mL streptomycin, 1 mM sodium pyruvate (11360-070, Gibco), 1% glutaMAX (35050-061, Gibco), and 1% non-essential amino acids (11140-050, Gibco). Patient F741 fibroblasts were kindly provided by Professor Hanna Mandel (Institute of Human Genetics, Galilee Medical Center, 22100 Nahariya, Israel)[13]. *Drp1*$^{+/+}$ and *Drp1*$^{-/-}$ MEFs were provided by Michael Ryan (Monash University, Melbourne, Victoria, AU). All cells were incubated at 37 °C under 5% CO$_2$ and 95% of humidified air. For lentivirus package, HEK293T cells maintained in DMEM with 10% HI-FBS, 100 U/mL penicillin, and 10 μg/mL streptomycin were co-transfected with pCMVdeltaR8.91, pCMV-MD.G and shRNA plasmids. Both shLacZ control (TRCN0000072224) and shNME3 were obtained from RNAi core, TRCN0000037747 with shRNA against 310-to-330 of NM_002513.2 as human *NME3*. After transfection for 48 and 72 h, supernatants containing lentivirus were filtered through PVDF membrane (pore size 0.45 μm, Millipore). For siRNA transfections, cells were seeded at 40%–50% confluency and the day after transfected with siRNA using Lipofectamine 2000 (Life Technologies) according to the manufacturer's protocol. siRNAs were purchased from Dharmacon: siControl (D-001210-02-20), siNME3 (L-006753-00-0005), siMUL1 (M-007062-02-0005 for human MUL1 and M-050675-00-0010 for mouse Mul1), siDRP1 (M-012092-01-0005), siFUNDC1 (M-018480-01-0005), siMARCH5 (M-007001-01-0005), and siRNF185 (M-007107-01-0005). The RNA extraction and RT-PCR were performed to validate the depletion of *NME3* according to published reference[38]. The primers for RT-PCR of *NME3*, *GAPDH*, *RNF185*, and *beta-actin* were shown in the Supplementary Table 1.

### Establishment of HA- and GFP-labeled endogenous NME3, MUL1 knockout, and NME3 knockout cell lines by CRISPR

EGFP (enhanced green fluorescent protein) and HA (human influenza hemagglutinin) inserted at the C-terminus of endogenous *NME3* by CRISPR was performed. The donor plasmids pUC19-hNME3-EGFP-KI and pUC19-NME3-HA-P2A-ZEO-KI containing sgRNA sequences (Supplementary Fig. S3c, d) were constructed and co-transfected with Cas9-mCherry to HeLa cells, respectively. After two days, cells were sorted by mCherry-positive fluorescence (excitation at 587 nm and emission at 610 nm), subsequently cells were seeded into 96-well for single-cell colony formation. The cell lysates from the colonies were examined by Western blotting for GFP and HA-labeling. DNA isolated from the positive colony was digested by restriction enzyme for confirm the knock-in position. To knockout *MUL1*, HeLa cells were transfected with lentiCRISPR v2-hMUL1 sgRNA2 plasmid with guide RNA sequence of human *MUL1* (5′-GTA CTC CGT GTA CCG GCA GA-3′). Two days after transfection, cells were selected with 1 μg/mL of puromycin to generate single-cell colonies. The colonies were analyzed by Western blotting and genome sequencing to confirm *MUL1* knockout. For knockout of human *NME3* gene, the lentiCRISPR v2-hNME3 sgRNA plasmids with guide RNA sequence of human *NME3* (5′-CTT CGC TAA CCT CTT CCC CG-3′) were used.

### Establishment of mt-Keima cell lines

To generate mt-Keima virus, pHAGE-mt-Keima plasmids were co-transfected with pCMV delta-8.91 and VSV-G using TurboFect

transfection reagent in 293T cells. 1% BSA in complete medium was added into media to increase the efficiency of viral titer. After 48 h of incubation, the media containing lentivirus was collected and filtered by 0.45 μm PVDF membrane. HeLa cells were infected by the virus filtrate containing 8 μg/mL polybrene. Mt-Keima-positive cells were sorted by flow cytometry by mt-Keima positive fluorescence (excitation: 560 nm; emission: 605 nm).

### Hypoxia condition

Cells were incubated in anaerobic incubator (SCI-tive Hypoxia Workstation, Ruskinn) which was purged with 0.5% O$_2$, 5% CO$_2$, and 94.5% N$_2$.

### Mitophagy analysis by mt-Keima

The mt-Keima is a mitochondrial matrix localized pH-indicator protein. In lived cells, mt-Keima fluorescence was measured using dual excitation ratiometric pH measurements with 488 nm (pH 7) and 561 nm (pH 4) lasers and 620 nm emission filters for neutral Keima (Mito) and acidic Keima (Lyso), respectively. The live dual excitation images were captured by inverted confocal fluorescence microscopy (Spinning Disc, Carl Zeiss) equipped with Yokogawa CSU-X1 microlens and a 63× objective lens under at 37 °C. Images of live cells were captured for 15 slices × 0.7 μm intervals and then performed a maximum intensity projection of a z-stack. The quantification of the percentage of mt-Keima in lysosome normalized by total mt-Keima in area was performed according to published reference[15].

### Autophagy flux

Cells were transfected with pMRX-IP-GFP-LC3-RFP-LC3DG (Addgene #84572). Cells stably expressing GFP-LC3-RFP-ΔLC3 were sorted by FACS. After starvation, these cells were harvested and subjected to flow cytometry analysis.

### Detection of histidine phosphorylation

To detect histidine phosphorylation, which is sensitive to acid and heat, all buffers were adjusted to pH 8.8 and heating was avoided. For in vitro kinase reaction, the recombinant protein of NME3-His were incubated in kinase buffer (50 mM Tris-HCl pH8.0, 5 mM MgCl$_2$, and 2 mM ATP) for 1 h at 37 °C. Half volume of samples was incubated for 10 min at 95 °C as a negative control for pHis. By performing Western blotting, both the stacking and separating gels were pH 8.8. Proteins were resolved at 90 V at 4 °C and transferred to PVDF membrane at 40 V for 16 h at 4 °C. Then the membrane was blocked in blocking buffer (0.1% BSA, 0.2× PBS) for 1 h at room temperature and incubated with antibodies against N1-Phosphohistidine (1-pHis) (1:2000, clone SC50-3, Millipore) for 1 h at room temperature which are diluted with blocking buffer containing 0.1% Tween-20. The membrane was washed 3 times for 10 min each with TBST (pH 8.8) at room temperature, followed by incubation with secondary antibody in blocking buffer with 0.1% Tween-20 and 0.01% SDS at room temperature for 1 h. Unprocessed images of Western blots are provided in the source data.

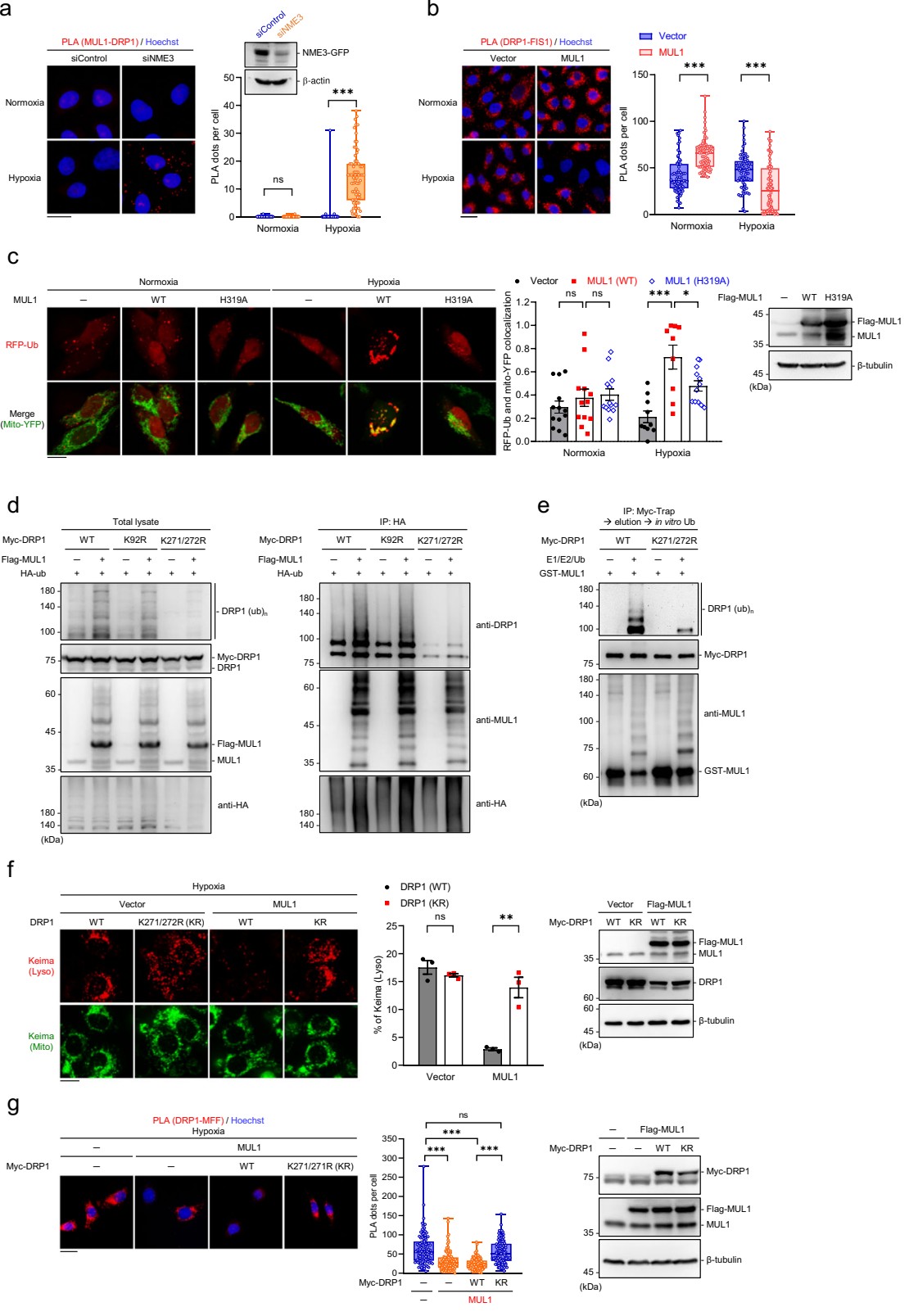

## Mice

*Nme3 H135Q* knock-in mutant mice were generated using CRISPR-Cas9 technique[41,42]. In brief, the *Nme3 H135Q* sgRNA (5′-CCG CAC AGG AAT GTA ATT CA-3′) was designed to target exon 5 of *Nme3* to generate Cas9-mediated double-strand break, and the antisense single-strand oligonucleotides (5′-GCT GTC CTC CCA GCA CAG AAG CTC GGC CTC ACG GAA CCA AAG AGC GAT TTC TCT GTG AGC ACT TTC CAC CGA

GTC GCT GCC CTG GAT AAC GTT CCT GTG CGG GAA TGG CAA GTG AGC AGG GGG AGG G-3′) was designed to introduce an artificial AclI restriction enzyme cutting site for genotyping and to change histidine residue 135 (codon CAT) to glutamine (CAG) based on mouse *Nme3* gene sequence on Ensembl (Transcript ID: ENSMUST00000024978.6). After zygote electroporation, two independent founders were selected for genotyping and sequencing.

**Fig. 6 | MUL1-mediated DRP1 ubiquitination disables mitophagy. a** HeLa cells transfected with siControl and siNME3 in normoxia and hypoxia for 24 h were analyzed by PLAs using MUL1 and DRP1 antibodies. Representative images are shown. Quantitation of the number of PLA dots per cell are presented by the box-and-whiskers plots ($n = 74$, 71, 81, and 69 cells) with Western blot of the samples. **b** Cells transfected with Flag-MUL1 or empty vector in normoxia and hypoxia for 24 h. Quantitation of PLA foci in each cell is shown ($n = 70$ cells). **c** Cells transfected with RFP-ubiquitin (Ub) and mitoYFP together with vector and MUL1 variants in normoxia and hypoxia overnight were analyzed by confocal microscopy. The representative images and quantification data ($n = 13$, 12, 12, 10, 9, and 12 cells) are shown. **d** HEK293T cells transfected with the indicated Myc-DRP variants with Flag-MUL1 and HA-ubiquitin. Cell lysates were immunoprecipitated using HA-beads in diluted Laemmli buffer and were analyzed by Western blot. **e** In vitro ubiquitination of Myc-DRP1 by GST-MUL1. Myc-DRP1 variants expressed in HEK293T cells were immunoprecipitated. The eluted Myc-DRP1 was incubated with GST-MUL1 for in vitro ubiquitination reaction and were analyzed by Western blot using DRP1 and MUL1 antibodies. **f** HeLa cells expressing mt-Keima transfected with Myc-DRP1 variants together with control vector or Flag-MUL1 were incubated in hypoxia for 24 h for mt-Keima analysis. The representative images, Western blot of the samples, and quantitation data ($n = 3$ independent experiments) are shown. **g** Cells transfected with MUL1 and Myc-DRP1 variants in hypoxia for 24 h were analyzed by PLA of DRP1-MFF interaction. Representative images, Western blot of the samples, and the number of PLA dots per cell in box-and-whiskers plots ($n = 100$ cells) are shown. Images are shown with scale bar, 20 μm. All box-and-whiskers plots show minima and maxima of all individual points with box ranged from 25th to 75th percentile. All quantified data are expressed as mean ± SEM. NS means no significance, *$p < 0.05$, **$p < 0.01$, ***$p < 0.001$; two-tailed $t$-test.

These transgenic mice were generated by the Transgenic Mouse Models Core Facility in National Taiwan University, and were bred and housed in a specific pathogen-free facility. The mice were backcrossed with C57BL/6J WT mice for three generations. Then, heterozygous *Nme3 H135Q* knock-in mutant mice (*Nme3 H135Q*^(m/+)) were intercrossed to have homozygous (*Nme3 H135Q*^(m/m)), heterozygous (*Nme3 H135Q*^(m/+)), and *Nme3* WT (*Nme3*^(+/+)) mice as littermate for experiments. For the genotyping of *Nme3 H135Q* knock-in mutant mice, the following primers were used for PCR followed by AclI digestion: forward (5′-CAC GTT CTT GGC AGT GAA GC-3′) and reverse (5′-CGA CTA GGT TGG GTT GAC CT-3′). Homozygous knock-in would result in two DNA bands (568 bp and 226 bp) after AclI digestion while heterozygous knock-in would show three bands (794 bp, 568 bp, and 226 bp). Wild-type mice would show just one DNA band (794 bp) after PCR and AclI digestion. The animal protocols were approved by the Institutional Animal Care and Use Committee (IACUC) of National Taiwan University (IACUC#20200030, March 1, 2020, to Feb 28, 2023, and IACUC #20201125, March/1/2021 to March/1/2024).

### Ledge test of mice
Mice were placed on the cage's ledge for ledge test based on published reference [43]. The phenotypes of mice walking along the edge of cage were recorded by video for analysis.

### Ischemia/reperfusion (I/R) mouse model and measurement of infarct size
The animal ischemia/reperfusion (I/R) protocol was approved by the Institutional Animal Care and Use Committee (IACUC) of National Taiwan University (#20201125). *Nme3* WT (*H135Q*^(+/+)) litter mate and *Nme3* homozygous *H135Q* knock-in mutant (*H135Q*^(m/m)) mice at 8–12 weeks old were subjected to anesthetization with Avertin (0.25 mg/kg, intraperitoneally) with ventilation through the tracheostomy using a rodent respirator for I/R as previously described[44,45]. Briefly, I/R experiment was conducted by midline sternotomy followed by snare occlusion near the left anterior descending artery. Myocardial I/R was achieved by tightening the snare for 20 min and then releasing it for 24 h reperfusion. Afterwards, the mice were sacrificed and the hearts were collected for measurements of weight and infarct size. To measure the infarct size, the ascending aortas from excised hearts were catheterized from the distal Valsalva to the sinus of Valsalva followed by retrograde perfusion with Evans blue (0.05%, E2129, Sigma-Aldrich) in the coronary arteries. The hearts were collected and frozen at −30 °C for 20 min and cut into 8 slices for further incubation in 0.9% sodium chloride buffer containing 1% of 2,3,5-triphenyl-tetrazolium chloride (TTC Red, T8877, Sigma-Aldrich) for 5 min at 37 °C. Then the sections were fixed with 4% paraformaldehyde for 15 min at room temperature. The infarct and the left ventricle area were measured using the planimetry method with Image J software. The infarct size was determined by dividing the infarct area by left ventricle area.

### Detection of endogenous mouse NME3 protein in cerebellums by mass spectrometry
Mitochondrial fractions were prepared from the homogenates of cerebellum tissues from mice by mitochondria isolation kit (130-096-946, Miltenyi Biotec) with mouse anti-Tom22 microbeads (130-127-693, Miltenyi Biotec) according to the vender's protocol, followed by SDS-PAGE separation. The gels spanning molecular weight region between 15 and 25 kDa were excised for in-gel trypsin digestion under aseptic conditions. Following reduction using final 5 mM Tris (2-carboxyethyl) phosphine hydrochloride (TCEP, cat# C4706, Sigma-Aldrich) and alkylation by final 10 mM iodoacetamide (37 °C, 30 min with agitation), samples were digested overnight at 37 °C in mass spectrometry grade trypsin, in 25 mM Triethylammonium bicarbonate. Peptides were extracted with 50% acetonitrile (ACN) with 5% formic acid (FA) and then 100% ACN dried via Speed-Vac. Samples were desalted by C18 Ziptip, dried via Speed-Vac, and redissolved by 0.1% FA for LC-MS/MS analysis. The mass spectrometry analysis was performed on a Thermo Fisher Scientific™ Orbitrap Fusion™ Tribrid™ mass spectrometer connected to an UltiMate™ 3000 RSLCnano System (Thermo Fisher Scientific, Bremen, Germany) equipped with a nanospray interface (Proxeon, Odense, Denmark). Peptides were loaded onto a 75 μm ID, 25 cm C18 BEH column (Waters, Milford, MA) packed with 1.7 μm particles with a pore size of 130 Å and were separated on a 70-min segmented gradient at a flow rate of 300 nL/min. Orbitrap survey MS1 scans of peptide precursors were acquired from m/z 350 to 1600 at 120 K resolution, with a target value of $2 \times 10^5$ ion count. The included charge states were 2–6 and the maximum injection time was 50 ms. Tandem MS was performed by isolating the precursor ions at a window of 1.6 Th in the quadrupole and fragmented by a higher-energy collisional dissociation (HCD) workflow at 30 collision energy. The MS2 fragments were detected simultaneously in the Orbitrap at a resolution setting of 30 K with a target value of $5 \times 10^4$ ion count. The maximum injection time was 54 ms. The dynamic exclusion duration was set to 15 s with 10 ppm tolerance around the selected precursor and its isotopes. Monoisotopic precursor selection was turned on. The raw data were processed using Proteome Discoverer 2.5 (Thermo Fisher Scientific), and peptide identification was performed by SEQUEST against the SwissProt database (v2022-10-12, total 17,108 sequences from Mus musculus). with a percolator (strict false discovery rate (FDR) of 0.01 and a relaxed FDR of 0.05). The protease was specified as trypsin with a maximum of 2 missing cleavage sites. Mass tolerance for precursor ion mass was 10 ppm with a fragment ion tolerance of 0.05 Da. Carbamidomethyl at cysteine was set as static modification, and oxidation at methionine, deamidation at asparagine or glutamine, acetyl at the protein N-terminus were selected as variable modifications. Peptides were considered to be identified if their individual ion score was higher than the identity score ($p < 0.05$). To evaluate the false discovery rate (<1%) in protein identification, a decoy database search against a randomized decoy database created by PD2.5 using identical search

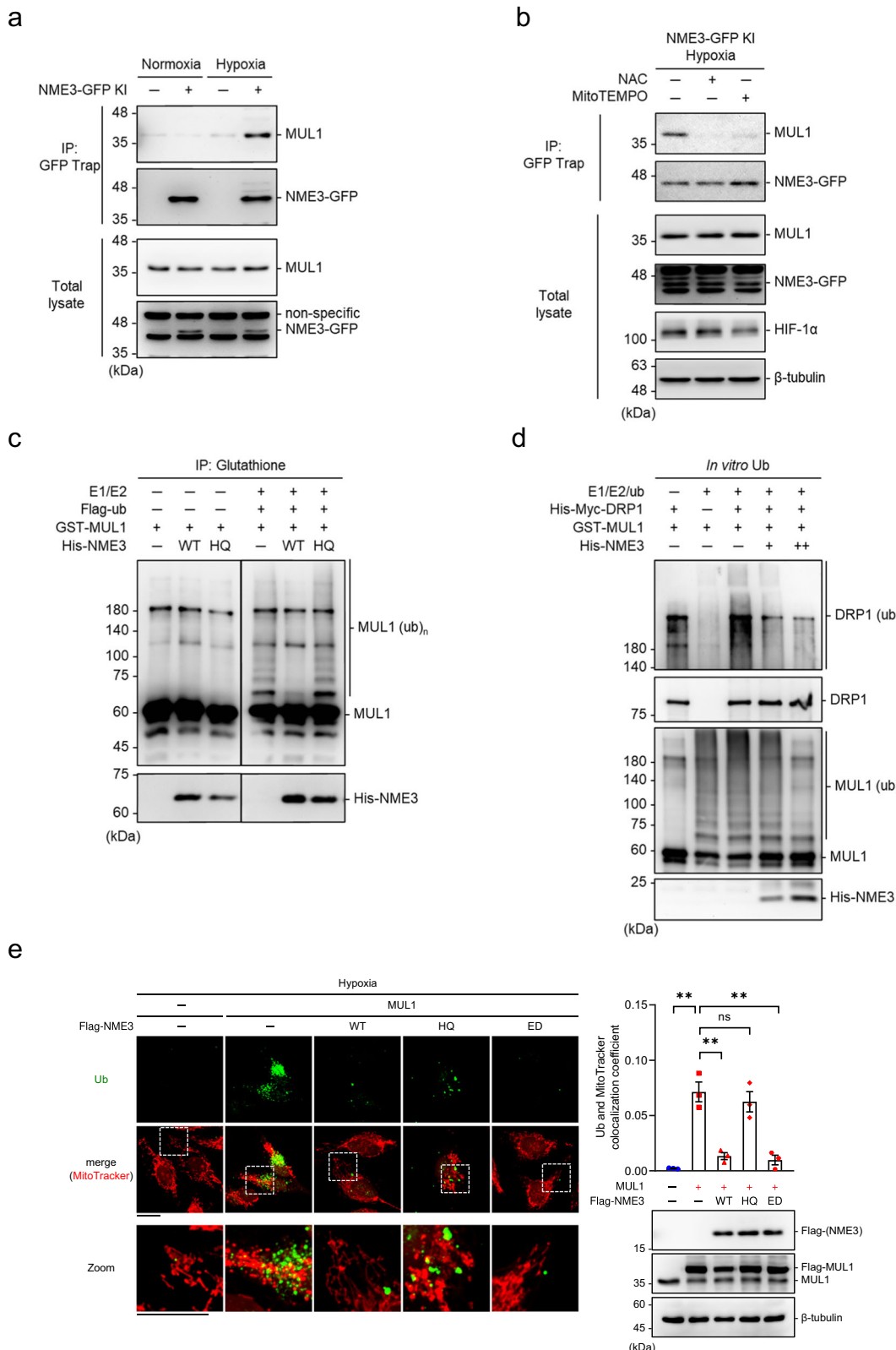

parameters and validation criteria was also performed. Peptide-spectrum matches (PSMs) with at least high confidence and a strict maximum parsimony principle (target FDR < 0.01) were applied for protein levels. Label-free quantification was performed using the peak area of each precursor ion calculated from extracted ion chromatogram during data processing using the Precursor Ions Area Detector node with mass precision 2 ppm. The abundance of identified protein was calculated from the top three of all unique and razor peptides in Peptide and Protein Quantifier node, and was used to calculate the relative protein abundance between experimental samples.

**Isolation of murine embryonic fibroblasts**

The MEFs with *Nme3 H135Q*[+/+] and *Nme3 H135Q*[m/m] genotypes were isolated according to a standard trypsin-mediated protocol. Briefly,

**Fig. 7 | NME3 interferes MUL1-mediated ubiquitination. a, b** Reactive oxygen species (ROS)-induced interaction of NME3 and MUL1. **a** HeLa cells with or without GFP knock-in that label endogenous NME3 were incubated in hypoxia (0.5% oxygen for 24 h) versus normoxia. Cell lysates were subjected to GPF-Trap pulldown for analyzing MUL1 association by Western blots. **b** In hypoxia for 24 h, cells were treated with 1 mM of N-acetylcysteine (NAC) and 20 μM of MitoTEMPO for NME3-GFP pulldown analysis. **c** Effect of NME3 on E3 ubiquitin activity. GST-MUL1 was incubated with WT and H135Q (HQ) of His-tagged NME3 proteins in the presence of ATP for 30 min, followed by glutathione beads pulldown. The beads were used for in vitro auto-ubiquitination reaction by addition of E1, E2, and Flag-ubiquitin for 30 min, and then the proteins were analyzed by Western blot using MUL1 and His antibodies. **d** Effect of NME3 on DRP1 in vitro ubiquitination by MUL1. Purified DRP1 was incubation with MUL1 in combination of WT NME3 for in vitro ubiquitination reaction. The reaction mixtures were analyzed by Western blot analysis. **e** HeLa cells were transfected with expression vector of MUL1 together with WT, HQ, and ED mutants of Flag-NME3 plasmids. After hypoxia for 24 h, cells were incubated with MitoTracker Red and fixed for IF staining of endogenous ubiquitin (Ub) followed by confocal microscopy analysis, scale bar, 20 μm. Colocalization coefficient of Ub and MitoTracker Red are shown as mean ± SEM (n = 3 independent experiments). NS means no significance, **p < 0.01; two-tailed t-test.

the pregnant mice with 13.5-day-old mouse embryos were sacrificed for dissecting the uterine horns, in which embryos were retrieved and transferred to Petri dishes containing PBS. After removing heads and visceral organs, embryos were washed extensively and minced. The dissected mouse embryos were incubated with 0.25% of trypsin-EDTA for 25–30 min for 5 times to generate a single-cell suspension. Then the MEFs were recovery in gelatin-coated culture dishes with culture medium (high glucose DMEM, 10% HI-FBS, 1% glutaMAX, 1% penicillin/streptomycin) for cell growth. The liver of the embryos was saved for DNA isolation and genotyping.

### Electron microscopy of mitochondrial morphology
For electron microscopy analysis, cells detached with 0.05% trypsin-EDTA were pelleted by centrifuging at $900 \times g$ for 5 min. The pellet was resuspended and fixed in 5% glutaraldehyde for 2 h at 4 °C and was post-fixed in 2% osmic acid for 2 h at room temperature, followed by dehydration, and embedded in Epon 812 resin (Polysciences). Ultrathin (70 nm) sections were stained with uranyl acetate and lead citrate and observed under a transmission electron microscope (H7100, Hitachi). Seventeen to twenty-seven fields with mitochondria were randomly photographed at an original magnification of ×50,000 and mitochondria (n > 50 in each group) were analyzed in a blinded setting. All mitochondria engulfed by membrane and total mitochondria were counted by Image-Pro PLUS software (Media Cybernetics).

### Mitochondria isolation
Mitochondria were isolated according to the manufacturer's protocol (mitochondrial isolation kit, #89874, ThermoFisher Scientific). Cells on a 10-cm culture dish were washed twice with ice-cold PBS and harvested in 500 μL of PBS. After centrifugation, cells were resuspended in 800 μL of reagent A containing 1% of protease inhibitor cocktail and 1% phosphatase inhibitor cocktail and were vortexed for 5 s. After incubation on ice for 2 min, cells were homogenized with Dounce Tissue Grinder with tight pestle for 60 strokes. Cell homogenates were added with equal volume of reagent C containing 1% of protease inhibitor cocktail and 1% phosphatase inhibitor cocktail, followed by centrifugation at $700 \times g$ for 10 min at 4 °C to remove nuclei and debris. The supernatants were pelleted by centrifuged at $3000 \times g$ for 15 min at 4 °C, followed by washing twice with reagent C to obtain mitochondrial enriched fraction.

### Immunofluorescence staining and microscopy analysis
Cells seeded on coverslips were treated with MitoTracker Red CMXRos (0.3 μM in cell culture medium, M7512, ThermoFisher Scientific) for 15 min at 37 °C, followed by PBS washed twice. Then cells were fixed with warm 4% paraformaldehyde at 37 °C for 20 min. After fixation, cells were washed twice with PBS and permeabilized with ice-cold TBST (50 mM Tris, 150 mM NaCl, and 0.3% Triton X-100) at room temperature for 10 min. Cells on coverslips were incubated with TBST containing the primary antibodies overnight, followed by washing and stained with secondary antibodies containing Hoechst 33342 (H1399, Invitrogen, 2 μg/mL) for 1 h at room temperature. After washing three times with TBST for 5 min, coverslips were mounted with mounting oil. The fluorescence images were acquired by using a fluorescence

microscopy (AxioObserver A1, Carl Zeiss) with AxioVision software (v4.8, Carl Zeiss) or a confocal fluorescence microscopy (LSM780, Carl Zeiss) equipped with ZEN software (Carl Zeiss, v2009). For measuring NME3-GFP signal in NME3-GFP knock-in HeLa, cells were treated with 100 nM of tetramethyl rhodamine ethyl ester perchlorate (TMRE, 87917, Sigma-Aldrich) for 10 min and then observe live cell fluorescence signal by using a confocal fluorescence microscopy (LSM780, Carl Zeiss) equipped with ZEN software (Carl Zeiss, v2009). The endogenous DRP1 and NME3-GFP, and ubiquitin signals on mitochondria were captured by using a super-resolution confocal fluorescence microscope (LSM880 with AiryScan detector, Carl Zeiss) equipped with ZEN software (Carl Zeiss, v2009).

### Proximity ligation assay (PLA)
Cells seeded on glass coverslips were washed with warm PBS twice and fixed with warm 4% paraformaldehyde at 37 °C for 20 min. After fixation, cells were washed twice with PBS and permeabilized with Tris-buffered saline (TBS, 50 mM Tris, 150 mM NaCl) with 0.3% Triton X-100 for 10 min. The coverslips were washed twice with TBST (TBS with 0.1% Triton X-100) and blocked with Duolink Blocking Solution containing Hoechst 33342 (H1399, Invitrogen, 2 μg/mL), Alexa Fluor 488 Phalloidin (A12379, ThermoFisher Scientific, 1:100 dilution) for 1 h in 37 °C. Then coverslips were incubated with incubated with the primary antibodies overnight at 4 °C. The antibodies for proximity ligation assays are: DRP1 (250× dilution, mouse antibody, ab56788, Abcam) and FIS1 (250× dilution, rabbit antibody, 10956-1-AP, Proteintech) for DRP1-FIS1 interaction; MFF (250× dilution, rabbit antibody, 10790-1-AP, Proteintech) for DRP1-MFF interaction; MUL1 (200× dilution, rabbit antibody ab209263, Abcam) and HA (200× dilution, mouse antibody, gtx628489, Genetex) for MUL1-NME3 interaction in NME3-HA knock-in cells; MUL1 (200× dilution, rabbit antibody ab209263, Abcam) and DRP1 (250× dilution, mouse antibody, ab56788, Abcam) for MUL1-DRP1 interaction in HeLa and MEFs. After washed twice with Duolink in situ wash buffer A (10 mM Tris, pH 7.4, 150 mM NaCl, 0.05% Tween) for 5 min, cells on coverslips were incubation with PLA anti-mouse PLUS probe (DUO92001, Sigma-Aldrich) and anti-rabbit MINUS probe (DUO92005, Sigma-Aldrich) for 1 h at 37 °C. After washing twice with Duolink in situ wash buffer A, cells were washed with TBST for 5 min twice and subjected to the ligation reaction for 30 min at 37 °C, followed by the amplification reaction (DUO92008, Sigma-Aldrich) for 100 min at 37 °C. After washing with PLA washing buffer B for 10 min, coverslip was mounted to the slide with mounting oil for microscopy analysis. Images were obtained by using fluorescence microscopy (Carl Zeiss, AxioObserver A1). AxioVision (Rel. 4.8, Carl Zeiss) and image J (1.52p, National Institutes of Health, USA) particles analysis was used for PLA foci quantification.

### Immunoprecipitation
Cells were harvested with IP buffer (50 mM Tris-HCl pH 7.4, 150 mM NaCl, 0.5% NP-40, 20 mM N-ethylmaleimide (NEM), and 1% protease inhibitor cocktail), followed by sonication and pre-clearance using 1 mg/mL BSA-treated Sepharose. Supernatant collected from centrifugation were incubated with antibody-conjugated agarose beads for at least 4 h at 4 °C with rotation. The samples were centrifuged at

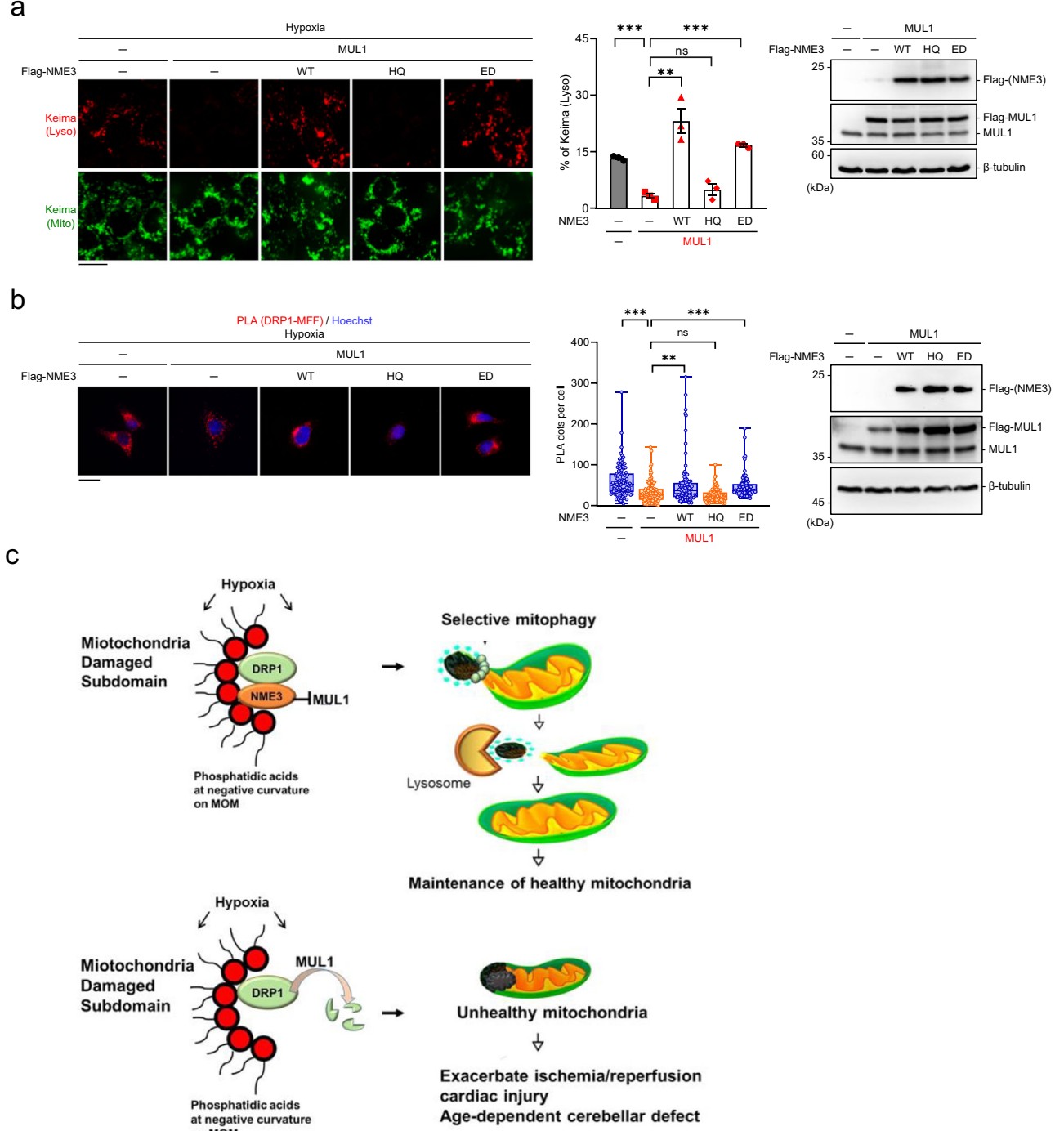

**Fig. 8 | Histidine phosphorylatable NME3 antagonizes MUL1 in mitophagy suppression. a–c** HeLa cells were transfected with expression vector of MUL1 together with WT, HQ, and ED mutants of Flag-NME3 plasmids. After hypoxia for 24 h, cells were analyzed for mitophagy and PLA of DRP1-FIS1 interaction. **a** Cells expressing mt-Keima were used for transfection. The representative images of mt-Keima fluorescence (*Left*), scale bar, 20 μm, quantification of percentages of lysosomal Keima of total Keima fluorescence in area (*Middle*) (*n* = 3 independent experiments), and Western blot analysis (*Right*) are shown. **b** Cells were fixed for PLA of DRP1-MFF interaction. The representative images (*Left*), scale bar, 20 μm, and quantification of the number of PLA dots per cell (*Middle*), and Western blot of the samples (*Right*) are shown. The quantified data are presented by the box-

and-whiskers plots (*n* = 100 cells), with minima and maxima of all individual points with box ranged from 25th to 75th percentile. All quantified data are presented as mean ± SEM. NS means no significance, **$p < 0.01$, ***$p < 0.001$; two-tailed *t*-test. **c** A proposed model. Hypoxia-induced PA formation on mitochondria promotes DRP1 interaction with WT NME3, which prevents the susceptibility of DRP1 to MUL1-mediated ubiquitination. This process maintains a sufficient amount of active DRP1 to segregate the damaged subdomain for selective mitophagy. Without histidine phosphorylatable NME3, DRP1 in dividing damaged domain is disabled by MUL1, thus increasing the accumulation of unhealthy mitochondria to exacerbate ischemia/reperfusion in cardiac injury and facilitate age-dependent cerebellar defect.

2500 × *g* for 2 min and washed the beads with IP buffer containing 0.1% NP-40. The following are the beads used for immunoprecipitation or pulldown analysis. Anti-Flag M2 affinity gel (A2220, Sigma-Aldrich), EZview Red Anti-HA affinity gel (E6779, Sigma-Aldrich), Glutathione Sepharose 4B (17-0756-01, GE healthcare), Ni-NTA Agarose (1018244, QIAGEN), and Protein G Sepharose 4 Fast Flow (GE17-0618-01, GE healthcare). Finally, the samples were suspended with 2× of Laemmli sample buffer and boiled at 95 °C for 3 min. All samples were analyzed by using Western blotting. For GFP trapping, cells were harvested with IP buffer containing 0.2% SDS. The supernatant was incubated with GFP-Trap Magnetic Agarose (gtma-20, ChromoTek) or Myc-Trap Magnetic Agarose (ytma-10, ChromoTek) for 1 h at 4 °C with rotation. Unprocessed images of Western blots are provided in the source data.

### In vitro ubiquitination reaction

The purified His-tagged NME3 (His-NME3) on Ni-NTA Agarose (1018244, QIAGEN) was incubated in a 20 µL reaction mixture containing 50 mM Tris-HCl pH 7.5, 5 mM $MgCl_2$, 2 mM sodium fluoride, 1 mM 1,4-dithiothreitol, and 2 mM ATP for 30 min, followed by the addition of 10 mM creatine phosphate, 0.5 µg creatine kinase, 1 µM ubiquitin aldehyde, 10 µg Flag-ubiquitin, 50 nM E1 (Sigma-Aldrich), 500 nM E2 (Sigma-Aldrich), and 2 µg GST-MUL1 for another 30 min at 37 °C. After brief centrifugation to remove Ni-NTA beads, the supernatants were collected followed by the addition of Laemmli sample buffer for Western blot analysis. For preparation of DRP1 used in MUL1-mediated in vitro ubiquitination reaction, HEK293T were transfected with Myc-DRP1 wild-type (WT) or K271/272R (KR) mutant overnight. These cells were harvested with IP buffer (50 mM Tris-HCl pH 7.4, 150 mM NaCl, 0.5% NP-40, 0.5 mM 1,4-dithiothreitol, and 1% protease inhibitor cocktail) and immunoprecipitated using Myc-Trap according to the manufacturer's protocol (Myc-Trap Magnetic Agarose, catalog number: ytma-10, ChromoTek). For Myc peptide elution, Myc-Trap beads containing Myc-DRP1 after washing were incubated with 30 µL of kinase reaction buffer (50 mM Tris-HCl pH 7.5, 5 mM $MgCl_2$, 2 mM sodium fluoride, 1 mM 1,4-dithiothreitol) and 1 µL of Myc peptide (yp-1, ChromoTek) for 15 min at room temperature. The eluents were collected for in vitro ubiquitination reaction by adding 1 mM ATP, 10 mM creatine phosphate, 0.5 µg creatine kinase, 10 µg Flag-ubiquitin, 50 nM E1 (Sigma-Aldrich), 500 nM E2 (Sigma-Aldrich), and 2 µg GST-MUL1 for 2 h at 37 °C. The proteins were analyzed by the addition equal volume of 2× Laemmli sample buffer for Western blot analysis.

### Statistics and reproducibility

The Scatter dot plots were generated by Prism (v8, GraphPad Software). All values are presented as mean ± standard error of mean (SEM). The box-and-whiskers plots show box from 25th to 75th percentiles and the whiskers ranged from minima to maxima. Statistical analysis of the results from at least three independent experiments was performed by 2-tailed unpaired Student's *t*-test for comparison of two groups. The *p*-value less than 0.05 was considered as statistically significant. All the Western blot analysis were repeated for at least 2 time with similar results. Numerical data with statistical results and the exact *p* values are provided in Supplementary Data File.

### Reporting summary

Further information on research design is available in the Nature Portfolio Reporting Summary linked to this article.

## Data availability

The mass spectrometry data have been deposited to the ProteomeXchange

Consortium via the PRIDE[46] partner repository with the dataset identifier PXD049224. The uncropped blots of Western blots of all

Figures and Supplementary Figs. are provided in Supplementary Data File. Source data are provided with this paper.

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

## Acknowledgements

We thank Hanna Mandel (Institute of Human Genetics, Galilee Medical Center, Israel) for F741 patient fibroblasts and Michael Ryan (Monash University) for *Drp1*[−/−] MEFs. We thank the National RNAi Core Facility at Academia Sinica for providing shRNA plasmids, the image core, and animal center at College of Medicine National Taiwan University for the technical assistance and the generation of knock-in mutation mice. We thank the Academia Sinica Common Mass Spectrometry Facilities for Proteomics and Protein Modification Analysis (Institute of Biological Chemistry, Academia Sinica), supported by Academia Sinica Core Facility and Innovative Instrument Project (AS-CFII-111-209). We thank Chuang-Rung Chang (National Tsing Hua University, Taiwan) and Ya-Wen Liu (National Taiwan University) for providing plasmids. This work is supported by grants from National Science and Technology Council, Taiwan NSCT 111-2320-B-002-088 and NSCT 111 –2326-B-002-022 to Z.F.C., NSCT111-2634-F-002-017 to Center of Precision Medicine at National Taiwan University, and a USPHS National Cancer Institute award to T.H. (CA242443). T.H. is the Renato Dulbecco Chair in Cancer Research and is a Frank and Else Schilling American Cancer Society Professor.

## Author contributions

Z.-F.C. conceived and designed the research. C.-W.C. performed all mitophagy experiments and data organization. C.S. analyzed generated a series of DRP1 mutants for ubiquitination experiments. C.-W.C. and C.S. performed PLA experiments. X.-R.H generated knock-in cell lines. Y.F. provided Nme3[−/−] and Nme3[+/+] MEFs. C.-Y.H. purified proteins and performed all biochemical experiments. T.C. did mice back-cross, breeding, and genotyping. X.C. performed mice ledge tests. C.-H.F. and C.-W.T. established the MUL1 and NME3 knockout HeLa cell lines. C.-W.C., T.C., Y.-W.T. and K.-C.Y. performed the mice I/R experiments. S.-T.H. and T.-Y.Y. performed and analyzed EM images. Y.-J.C. performed mass spectrometric analysis of NME3 in mouse cerebellum. T.H. provided suggestions in histidine phosphorylation, data interpretation, and paper writing. Z.-F.C. and C.-W.C. wrote the paper. All authors have approved the submitted version. All authors have read and agreed to the published version of manuscript.

## Competing interests

T.H. holds a patent describing the generation and use of phosphohistidine-specific monoclonal antibodies. The remaining authors declare no competing interests.
