## [Peer Review File · Nature Communications]

NME3 is a gatekeeper for DRP1-dependent mitophagy in hypoxiaREVIEWER COMMENTS

Reviewer #1 (Remarks to the Author):

The authors show that NME3 prevents Drp1 from MUL1-mediated degradation of Drp1 during hypoxia, thereby allowing Drp1 to mediate mitophagy.

The study shows the novel function controlling the level of Drp1 during hypoxia through NME3, a member of the NDPK family utilizing phosphorylated histidine.

General:

Besides the protective effect of endogenous NME3 against Drp1 degradation, its functional significance toward the mechanism of mitophagy remains to be clarified. The authors suggest that the effect of NME3 upon MUR1-Drp1 is independent of the NDPK function. Although the identification of the un-autophosphorylatable histidine mutation (H135Q), as loss of NME3 function mutation (not protecting Drp1) is interesting, its significance is very unclear to this reviewer. Is H135 autophosphorylation regulated during hypoxia and involved in mitophagy? Can H135 auto-phosphorylation dissociate from NDPK function: Is there any condition in which H135 phosphorylation dissociates from the NDPK function of NME3?

The authors show that interaction between NME3 and MUR1 occurs only in the presence of hypoxia and that this interaction is inhibited by H135Q in an NDPK function-independent manner. Furthermore, MUR1 rather increases Drp1-Fis1 interaction at baseline. These observations are interesting, but the study fails to provide a mechanistic explanation.

The Ubiquitin ligase function of MUL1 is shown only in overexpression- or test-tube conditions.

Mitophagy is one of many functions of Drp1. The authors should clarify the mechanistic (or direct) involvement of Drp1 in mitophagy during hypoxia. The effect of Drp1 upon mitophagy shown here could be secondary to the effect of Drp1 upon mitochondrial function in general.

Specific:

The authors should provide the rationale to investigate the effect of H135Q mutation. It appears too sudden.

The study requires quantification in many data presented, including Fig 1bc.

Many experiments show data from only hypoxic conditions. It would be ideal to show normoxic data since the MUR1 has opposite effects upon DRP1-Fis1 interaction between normoxia and hypoxia.

In Figure 3,d, why is the number of PLA (Drp1-Mff) dots very different between siControl and WT MEF?

In Figure 4b, show the purity of mitochondria fraction (the lack of contamination from other fractions). In this experiment, why is the level of Drp1 higher in m/m samples?

The labeling of Figure 4d, is confusing. Is endogenous MUL1 probed in this case? Why are there no baseline MUL1 levels in the western blot?

In page 10, lines 13-14, please explain why the data suggest the potential function of NME3 in GTP loading is not essential.

Figure S6 does not show the ROS-dependent NME3 and MUL1 interaction.

Reviewer #2 (Remarks to the Author):

The manuscript by Chang group showed that NME3, a member of the nucleoside diphosphate kinase family, regulates mitophagy through its interaction with Mul1 to inhibit its E3 ligase activity. Subsequently, such interaction affects the ubiquitination of DRP1, leading to mitochondrial fission for mitophagy. It is reported that Mul1 functions as a positive mitophagy regulator, but here the authors found that Mul1 could negatively regulate mitophagy in hypoxia. The authors need to adequately discuss

this issue. It would be interesting to show if the phosphorylation status of NME3 regulate Mul1 activity and NME3 affects DRP1 phosphorylation? Other concerns include:

1. it would be interesting to check if knockdown of NME3 decrease the protein level of Drp1, and increase its ubiquitination especially under hypoxia?

2. Fig 6c: it is better to measure if NME3 could directly modulate DRP1 ubiquitination by Mul1 by this in vitro system.

- 4 Mitophagy should also be verified by western blotting of mitochondrial membrane and matrix proteins, in addition to MitoKeima.

Point-to-point responses:

Reviewer #1 (Remarks to the Author):

The authors show that NME3 prevents Drp1 from MUL1-mediated degradation of Drp1 during hypoxia, thereby allowing Drp1 to mediate mitophagy.

The study shows the novel function controlling the level of Drp1 during hypoxia through NME3, a member of the NDPK family utilizing phosphorylated histidine.
General:

Besides the protective effect of endogenous NME3 against Drp1 degradation, its functional significance toward the mechanism of mitophagy remains to be clarified.

RE: We totally agree with this point. In this revised version, we provide additional evidence to increase the clarity of the mechanism. Our new data demonstrate that hypoxia treatment promotes the complex formation of NME3 with DRP1 for mitophagy in a phosphatidic acid (PA)-dependent manner. These new data are shown in Fig. 3 a-h. The consideration of PA in the role of NME3 in hypoxia-induced mitophagy is based on our recent finding that NME3 is a PA binding protein published in *J Cell Biol.* 2023; 222: e202301091, which has been cited in the spotlight in the same issue of *JCB* and *Trends Cell Biol.* Since DRP1 also has been shown to have PA binding activity (*Mol Cell.* 2016; 63: 1034-1043), we speculated that hypoxia treatment might induce site-specific interaction between NME3 and DRP1, and PA might be a critical mediator. Data shown in Figure 3 of this revised version support this hypothesis. This site-specific interaction provides DRP1 a microenvironment to proceed the segregation process for mitophagy. Without NME3, DRP1 becomes susceptible to MUL1-mediated degradation (Fig. 8c).

- 1. Fig. 3a-e showed that hypoxia treatment increased PA on mitochondria and NME3 interaction with DRP1, which were reversed by expressing active lipin that converted PA to diacylglycerol. Thus, hypoxia-induced NME3-DRP1 interaction is PA-dependent, and hypoxia treatment indeed increases PA on mitochondria.**
- 2. In Fig. 3f, we showed that hypoxia-induced mitophagy requires PA formation.**
- 3. To strengthen the importance of PA binding of NME3 in mediating DRP1-dependent mitophagy. We replaced the N-terminal region of NME3 that confers PA binding and MOM-localized function with the TOM20 transmembrane sequence. So that MOM-localized NME3 mutant defective in PA binding is generated. We then compared the expression of WT-NME3 with TOM20-N Δ -NME3-PA binding mutant in restoring hypoxia-induced mitophagy**

in NME3-deficient cells. The results were very clear that unlike WT-NME3, the expression of MOM-NME3-PA binding mutant was unable to restore mitophagy. Thus, PA binding but not MOM-localization is essential for NME3's role in mitophagy.

4. We proposed a model in which hypoxia treatment increases PA formation, which provide a microenvironment for active NME3 interaction with DRP1. This process is critical for DRP1 function in selective mitophagy (Fig. 3i)

The authors suggest that the effect of NME3 upon MUR1-Drp1 is independent of the NDPK function. Although the identification of the un-autophosphorylatable histidine mutation (H135Q), as loss of NME3 function mutation (not protecting Drp1) is interesting, its significance is very unclear to this reviewer. Is H135 autophosphorylation regulated during hypoxia and involved in mitophagy?

RE: Thank you for this important question. The endogenous level of NME3 is very low not only due to low transcript level but also very short half-life (60-80 min). NME1 and 2 are very abundant NDPK isoforms in the cytosol and nucleus and they are well known histidine kinase. Because of instability of 1-pHis signal and very low level of endogenous, we were unable to specify the change in histidine phosphorylation of endogenous NME3 during hypoxia.

Can H135 auto-phosphorylation dissociate from NDPK function: Is there any condition in which H135 phosphorylation dissociates from the NDPK function of NME3?

RE: Data from Fig. 1d, e and Supplementary Fig. S1b-c showed that expression of WT or ED mutant but not HQ mutant of NME3 restored mitophagy signal in NME3-defective cells. We have previously shown that E40 and E46 residues mediate the interface interaction of NME3 subunits. Mutation of E40/E46 to D40/D46 (ED mutant) interferes oligomerization of NME3. The recombinant protein of ED mutant has very low NDPK activity but retains histidine 135 phosphorylation (Fig. 1f, g). Overexpression of ED mutant faithfully restored hypoxia-induced mitophagy. Therefore, NDPK activity of NME3 is unlikely involved in DRP1-mediated mitophagy. Moreover, in this study, we found that MUL1 overexpression increased ubiquitination signal associated with mitochondria and suppressed mitophagy in hypoxia condition, both of which were reversed by WT- and ED mutant of NME3, but not H135Q mutant (Fig. 7e and 8a-b). Given very low NDPK activity of ED mutant, these data suggest that the involvement of NME3 in these processes is uncoupled with its NDPK function. Therefore, we used the term of histidine 135 phosphorylatable NME3 and hope the clarity is improved. It should be mentioned

that in our laboratory, we have found that MUL1 protein after incubation with WT- but not H135Q mutant NME3 acquired 3-pHis phosphorylation, which could be inhibitory for ubiquitin accessibility of MUL1.

Figure 1. *In vitro* analysis of MUL1 phosphorylation by NME3.

(Left) Purified recombinant proteins of wild-type (WT) and catalytic-dead H319A mutant of GST-MUL1 were incubated with NME3-His in the presence of ATP for 30 min. The mixtures were divided with and without 95°C treatments followed by Western blot with antibodies of 3-pHis, 1-pHis, GST, and His.

(Right) Structural model for the 3-pHis-induced alteration of the RING-ubiquitin interface. The upper panel shows that structural perturbations caused by the YASARA energy minimization procedure are negligible in the absence of 3-pHis modification. The original crystal structure (PDB: 4AP4) and energy-minimized structure are in grey and cyan, respectively. The bottom panel shows a significant movement of the zinc ion and pronounced structural changes of the RING-ubiquitin contacts are observed when the same energy minimization procedure was conducted upon N-3 phosphorylation of the zinc-coordinating H225. The energy-minimized structures with and without the placement of phosphate at N3 position of H225 are in green and cyan, respectively. Note that H225 of RNF4 is equivalent to H319 in MUL1.

Because of very unstable nature of 3-pHis and the reproducibility problem of data, we did not pursue further. However, WT-NME3 did suppress MUL1 ubiquitination activity *in vitro* (Fig. 7c, d). In the present context, many questions remain open.

The authors show that interaction between NME3 and MUR1 occurs only in the presence of hypoxia and that this interaction is inhibited by H135Q in an NDPK function-independent manner. Furthermore, MUR1 rather increases Drp1-Fis1 interaction at baseline. These observations are interesting, but the study fails to

provide a mechanistic explanation.

RE: Data from the overexpression and co-immunoprecipitation experiment, we know that WT or H135Q NME3 forms a complex with MUL1.

Figure 2. Co-immunoprecipitation of overexpressed WT and H135Q-NME3-GFP with Flag-MUL1. HEK 293T cells were transfected with NME3-GFP variants with Flag-MUL1. Cell lysates were immunoprecipitated with Flag antibody, followed by Western blot analysis.

However, at endogenous level, we did not find their stable complex formation unless in the condition of hypoxia (Fig. 7a), which was abolished by mitoTempo treatment (Fig. 7b). This indicates that hypoxia-induced oxidative environment increases their interaction affinity. We assumed that the interaction of NME3 with MUL1, a cysteine-rich protein, is increased in response to ROS. We did *in vitro* pulldown assay of GST-MUL1 (full-length) with His-NME3 and found the increase of NME3 pulldown in the presence of increasing H₂O₂.

Figure 3. *In vitro* analysis of MUL1-NME3 interaction in response to hydrogen peroxide (H₂O₂) pre-treatments. Purified GST-MUL1 was treated with H₂O₂ for 30 min followed by addition of increasing amounts of purified NME3-His proteins for 30 min. Afterwards, glutathione beads were added and mixed. The GST-MUL1 pulldown and unbound supernatant were analyzed by Western blot using antibodies against His-tag and MUL1.

Since we were unable to define the cysteine residues responsible for ROS-sensitive interaction, we did not add the data in the revised manuscript. It is possible that *in vivo* condition, other mitochondrial factor sensitive to ROS increases their proximity for interaction. Again, we think the involvement of microenvironment on ROS-damaged mitochondria, for instance, PA, or other ROS-sensitive protein, might affect their interaction affinity.

It is interesting that we found opposite effect of MUL1 overexpression on DRP1-FIS1 interaction in normoxia and hypoxia (Fig. 6b). As noted, MUL1 overexpression in normoxia did not promote ubiquitin signal associated with mitochondria, which was also opposite to very prominent ubiquitin/mitochondria signal seen in hypoxia (Fig. 6c). Moreover, we found that the increase of mitochondrial ubiquitination signal in hypoxia was reduced by WT- and ED- but not H135Q NME3 (Fig. 7e). Therefore, factor(s) such as ROS generated from hypoxia stress might shift MUL1 function directly or indirectly to ubiquitinate proteins, including DRP1, on mitochondria (Fig. 5e). In normoxia, MUL1 overexpression might stabilize DRP1 by Sumoylation in apoptotic cells as suggested by the studies from McBride's group. We discussed this in the revised manuscript.

The Ubiquitin ligase function of MUL1 is shown only in overexpression- or test-tube conditions.

RE: In this revised manuscript, we provide evidence of MUL1-mediated ubiquitination by knockdown and knockout experiments. First, we showed that mitochondria-associated ubiquitination signal in NME3 knockdown cells was markedly increased by hypoxia (Supplementary Fig. S7a). However, MUL KO prevented hypoxia-induced mitochondrial ubiquitination signal in NME3 knockdown cells (Fig. S7b). Secondly, for DRP1, we used super-resolution microscopy to show that the colocalization of endogenous DRP1 with Ub signal in NME3-defective cells was abolished by MUL1 depletion (Fig. 5e).

Mitophagy is one of many functions of Drp1. The authors should clarify the mechanistic (or direct) involvement of Drp1 in mitophagy during hypoxia. The effect of Drp1 upon mitophagy shown here could be secondary to the effect of Drp1 upon mitochondrial function in general.

RE: The restoration of mitophagy in NME3-defective cells by DRP1 overexpression could be simply due to excess amount of DRP1 by overexpression that can compensate the loss of active DRP1 from MUL1 susceptibility (Fig. 4d). In this revised manuscript, we added the data showing that DRP1 overexpression in NME3-defective cells did restore the interaction between DRP1 and FIS1 (Fig. 4e),

thereby capable of driving the segregation process for mitophagy.

In addition, this revised manuscript also provides new data by super-resolution microscopy to show that hypoxia-induced LC3B puncta are highly associated with NME3/DRP1 co-localized signal (Supplementary Fig. S3e).

Data described by Kleele T *et al.* (*Nature*. 2021; 593: 435-439) showed that DRP1 division via FIS1 at peripheral region is for mitophagy in normal condition. The paper by McBride's group (*Nat Cell Biol.* 2021; 23: 1271-1286) showed that DRP1 is required for the formation of mitochondria-derived vesicles (MDVs), which are then degraded by lysosome. Of note, NME3 is present in their proteome analysis of MDVs. After hypoxia treatment, mitochondria are damaged by ROS. The segregation of damaged subdomain requires DRP1 interaction with fission receptor. Our data also showed that DRP1 is required for hypoxia-induced mitophagy (Fig. S3a, b). In this study, we found that DRP1/FIS1 and DRP1/MFF interaction both are reduced in NME3 KD or H135Q mutant cells, indicating that DRP1-mediated division via these two receptors are suppressed due to NME3 defect, thereby impairing the segregation process for mitophagy.

Specific:

The authors should provide the rationale to investigate the effect of H135Q mutation. It appears too sudden.

RE: In our previous studies, we showed that catalytic H135 is not required for NME3-mediated mitochondrial tethering for promoting mitochondrial fusion process. However, glucose starvation causes cell death of NME3-deficient patient cells, which is rescued by WT-NME3 but not H135Q NME3. This indicates that the catalytic H135 site is still very important in the stressed condition by exerting other function. We explained why we test the physiological function of catalytic function of NME3 in in the introduction of this revised manuscript.

The study requires quantification in many data presented, including Fig 1bc.

RE: We add the quantification data in Fig. 1b, c.

Many experiments show data from only hypoxic conditions. It would be ideal to show normoxic data since the MUR1 has opposite effects upon DRP1-Fis1 interaction between normoxia and hypoxia.

RE: In this manuscript, Fig. 3a, 3b, 3d, 4a, 4b, 6a-c, 7a, S4a, S5a, S6a, and S7a all show normoxia and hypoxia comparison data.

In Figure 3,d, why is the number of PLA (Drp1-Mff) dots very different between siControl and WT MEF?

RE: The difference of DRP1-MFF PLA signal in siControl HeLa and WT MEFs are very different in normoxia, which might be because they are different cell types and might have different basal level of MFF or modification, or the level of other fission factors available for interaction with DRP1 in normoxia. To avoid too much expansion of results, we moved most of MEFs data in Supplementary Figures.

In Fig. 4b, show the purity of mitochondria fraction (the lack of contamination from other fractions). In this experiment, why is the level of Drp1 higher in m/m samples?

RE: Agree with this point. We did another preparation as shown below, in which similar level of Drp1 in the mitochondrial fraction with higher amounts of ubiquitinated proteins in the mitochondria-enriched fraction in H135Q MEF.

Since we already showed that mitochondrial ubiquitin signal markedly increased in H135Q MEFs in Supplementary Fig. S5a and siNME3 transfection (Fig. S7a) in hypoxia not normoxia, the WB was removed to prevent redundancy.

The labeling of Figure 4d, is confusing. Is endogenous MUL1 probed in this case? Why are there no baseline MUL1 levels in the western blot?

RE: We are very sorry about this. It was probed by Flag antibody because of the transfection with Flag-MUL1(H319A). The data with corrected labeling of Flag-MUL1 was moved to Supplementary Fig. S5c in the revised manuscript.

In page 10, lines 13-14, please explain why the data suggest the potential function of NME3 in GTP loading is not essential.

RE: Thanks for raising this important question. In this revised manuscript page 14

and 19, we explain why we think that the GTP-loading of DRP1 is unnecessarily dependent on the association with NME3, because the expression of NME3-ED mutant defective in NDPK function, DRP1 overexpression, and MUL1 depletion all restore DRP1 interaction with fission receptor and hypoxia-induced mitophagy in NME3-defective cells.

Figure S6 does not show the ROS-dependent NME3 and MUL1 interaction.

RE: We are very sorry for this mistake in the caption. The Supplementary Fig. S3-S7 have been reorganized.

Reviewer #2 (Remarks to the Author):

The manuscript by Chang group showed that NME3, a member of the nucleoside diphosphate kinase family, regulates mitophagy through its interaction with Mul1 to inhibit its E3 ligase activity. Subsequently, such interaction affects the ubiquitination of DRP1, leading to mitochondrial fission for mitophagy. It is reported that Mul1 functions as a positive mitophagy regulator, but here the authors found that Mul1 could negatively regulate mitophagy in hypoxia. The authors need to adequately discuss this issue. It would be interesting to show if the phosphorylation status of NME3 regulate Mul1 activity and NME3 affects DRP1 phosphorylation? Other concerns include:

RE: We totally agree with this scientific comment. In this revised manuscript, we showed that the opposite effect of MUL1 overexpression on DRP1-FIS1 interaction in normoxia and hypoxia (Fig. 6b). As noted, MUL1 overexpression in normoxia did not promote ubiquitin signal associated with mitochondria, which was also opposite to very prominent ubiquitin/mitochondria signal seen in hypoxia (Fig. 6c). We showed that mitochondria-associated ubiquitination signal in NME3 knockdown cells was markedly increased by hypoxia (Supplementary Fig. S7a). However, in MUL KO cells, NME3 knockdown no longer caused hypoxia-induced mitochondrial ubiquitination signal (Fig. S7b). Moreover, we found that the increase of mitochondrial ubiquitination signal in hypoxia was reduced by WT- and ED- but not H135Q NME3 (Fig. 7e). For DRP1, we used super-resolution microscopy to show the colocalization of endogenous DRP1 and Ub signal in NME3-defective cells, which was abolished by MUL1 depletion (Fig. 5e). We also showed that the amount of pS616-DRP1 was reduced in NME3-defective cells (Fig. 4f, and Fig. S5f). MUL1 knockout or knockdown increased the level of pS616-DRP1 in these NME3-defective cells (Fig. 5d, and Fig. S5f). According to these data, we assumed that factor(s) such as ROS generated from stress might shift MUL1 function to ubiquitination of proteins, including DRP1, on mitochondria (Fig. 5e). In normoxia, MUL1 overexpression might stabilize DRP1 by Sumoylation as suggested by the studies from McBride's group. We discussed this in page 14 and 20 of this revised manuscript.

In addition, in this revised manuscript, we add new data in Fig. 3 to demonstrate that hypoxia treatment increases PA formation, which provide a microenvironment for active NME3 interaction with DRP1. This process is required for selective mitophagy. In Fig. 3i, we provide a model to depict how hypoxia-induced PA causes site-specific interaction of NME3 and DRP1 for mitophagy. In Fig. 8c, we

draw a model to explain that hypoxia-induced NME3-DRP1 complex at PA sites on mitochondria prevents DRP1 susceptibility to MUL-mediated ubiquitination. Without NME3, active DRP1 is susceptible to MUL1-mediated degradation, thereby impairing the segregation process required for mitophagy.

1. it would be interesting to check if knockdown of NME3 decrease the protein level of Drp1, and increase its ubiquitination especially under hypoxia?

RE: Thank for this valuable suggestion. In this revised manuscript, we showed that the amount of pS616-DRP1 was reduced in NME3-defective cells (Fig. 4f, and Fig. S5f). MUL1 knockout or knockdown increased the level of pS616-DRP1 in these NME3-defective cells (Fig.5d, and Fig. S5f).

2. Fig 6c: it is better to measure if NME3 could directly modulate DRP1 ubiquitination by Mul1 by this in vitro system.

RE: Thank you for this valuable suggestion. For this experiment, we purified DRP1 protein and incubation with GST-MUL (full length) in the absence and presence of NME3. The results showed that DRP1 ubiquitination by MUL1 is suppressed by NME3 (Fig. 7d).

4 Mitophagy should also be verified by western blotting of mitochondrial membrane and matrix proteins, in addition to Mitokeima.

RE: In this revised manuscript, we showed that the reduction of Vdac1 in WT MEFs but not in H135Q MEFs after hypoxia 2-3 days. Probably, it is a selective mitophagy under hypoxia treatment. We did not find clear reduction of many other mitochondrial proteins. In HeLa, we did not find difference in VDAC1. Again, different cell types might eliminate different subsets of mitochondrial proteins in selective-mitophagy.

REVIEWER COMMENTS

Reviewer #1 (Remarks to the Author):

The authors have addressed many of my concerns and the paper has improved significantly. However, this paper shows many data whose implication is unclear, which unnecessarily makes the whole story more complicated and puzzling. The interpretation of some results is unclear.

The authors could have stated a clear rationale for studying PA binding in Figure 3. It seems to me that this Figure is side-tracked since it merely showed the properties of hypoxia-induced mitophagy in these cells and did not add much to the story regarding the effect of NME3 upon MUL1-Drp1 regulation.

It is unclear to this reviewer why the authors show the Drp1-Fis1 interaction in some cases and the Drp1-Mff interaction in others.

In Figure 4d, the authors should include control si data to compare with siNME3 data. In this experiment, the rescue by Drp1 is only modest (compared to Figure 1d). Please comment.

In Figure 4f, please comment on the underlying mechanism as to why pDrp1/Drp1 is affected by the lack of NME3.

Reviewer #2 (Remarks to the Author):

the authors have addressed my comments.

Point-to-point response to reviewer #1

The authors have addressed many of my concerns and the paper has improved significantly. However, this paper shows many data whose implication is unclear, which unnecessarily makes the whole story more complicated and puzzling. The interpretation of some results is unclear.

RE: We are very sorry that the previous revision was still not clear enough. We hope this second revision clearer in the scenario and hypothesis by adding new data and more explanation in our proposed model and discussion.

The authors could have stated a clear rationale for studying PA binding in Figure 3. It seems to me that this Figure is side-tracked since it merely showed the properties of hypoxia-induced mitophagy in these cells and did not add much to the story regarding the effect of NME3 upon MUL1-Drp1 regulation.

RE: We know that data of figure 3 are not within the scope of the reviewers' concerns. However, we think that we should explain why NME3 regulates DRP1 in hypoxia but not normoxia. Since endogenous NME3 does not interact with DRP1 in normoxia unless hypoxia treatment, the mechanism delineating this interaction change should add the clarity why NME3 becomes a regulator of DRP1 at hypoxia-induced mitochondrial damage sites. Without this site-specific interaction with NME3, DRP1 at damage sites becomes susceptible to MUL1-mediated ubiquitination. Because we already published biochemical data showing that the N-terminal region of NME3 binds to phosphatidic acid in *J Cell Biol.* 2023; 222: e202301091, spotlight in the same issue of *J Cell Biol.* 2023;222:e202309037 and *Trends Cell Biol.* 2023;33:1005-1006), adding the results of Figure 3 should further highlight the theme that NME3 senses PA lipid signal on mitochondrial damage sites to exert its gatekeeper function by protecting DRP1 from MUL1. We thought this would strengthen the hypothesis with clarity and novelty. The effect of NME3 upon MUL1/DRP1 regulation was mainly demonstrated by MUL1 knockdown and knockout experiments in Fig. 5 and Fig. S5, and MUL1-mediated *in vitro* ubiquitination of DRP1 and mitochondrial ubiquitination in Fig. 7 and Fig. S7.

It is unclear to this reviewer why the authors show the Drp1-Fis1 interaction in some cases and the Drp1-Mff interaction in others.

RE: In this revision, we added the data of DRP1-MFF interaction in Fig. 4d and g, Fig. 5c, and Fig. S6b. All PLA signals of DRP1-MFF interaction were similar to what we observed for DRP1-FIS1.

In Figure 4d, the authors should include control si data to compare with siNME3 data. In this experiment, the rescue by Drp1 is only modest (compared to Figure 1d). Please comment.

RE: In this revision, we added siControl data in Fig. 4e. The data clearly showed that DRP1 overexpression only promoted mitophagy signal in NME3 knockdown but not the control cells, suggesting DRP1 becomes a limiting factor to affect mitophagy in NME3-deficient cells. Thank you for pointing out that DRP1 overexpression only partially rescued mitophagy signal in HeLa depleted of NME3. In page 12, we explained that in hypoxia, other factor important for mitophagy process probably is also partly regulated by NME3 in HeLa. Therefore, DRP1 overexpression was unable to fully rescue mitophagy even though the DRP1-MFF and DRP1-FIS1 interaction were fully restored.

In Figure 4f, please comment on the underlying mechanism as to why pDrp1/Drp1 is affected by the lack of NME3.

RE: In page 12-13, we explained the meaning of pDRP1/total DRP1 change. Since an excess amount of DRP1 by overexpression is capable of promoting mitophagy in the cells depleted of NME3 but not control cells, it indicates that DRP1 becomes a limiting factor for mitophagy in NME3-defective cells. Since the total amount of DRP1 is not decreased in NME3-defective cells, we then analyzed the amount of active form of DRP1, indicated by pS616, by Western blot analysis in mitochondria-enriched fractions. The result showed that the amount of active DRP1 in the mitochondrial fractions was particularly affected by NME3 knockdown (Fig. 4h), suggesting the importance of NME3 in maintaining a sufficient amount of active form of DRP1 on mitochondria. Without NME3, ectopic overexpression of DRP1 compensates the loss of active form of DRP1 to promote the dividing step for mitophagy.

Since hypoxia-induced NME3 interaction with DRP1 and mitophagy are both PA-dependent, this implies that NME3/DRP1 complex present in hypoxia-induced PA-enriched microenvironment on mitochondrial outer membrane (MOM) is critical for maintaining a sufficient amount of active form of DRP1 for the subsequent segregation. Our data shown in Fig. 7 and Fig. 8b provide evidence that the presence of NME3 protects DRP1 from MUL1-mediated ubiquitination to proceed fission for mitophagy. According to these data, we proposed a model (page 17-18), in which active form of DRP1 in complex with NME3 at hypoxia-induced PA lipid sites on MOM, is not susceptible to MUL1-mediated ubiquitination, thus enabling segregation division. Without histidine phosphorylatable NME3, DRP1 binding at the PA site is no longer protected and becomes susceptible to MUL1-mediated ubiquitination. The loss of active DRP1 at damage sites by MUL1-mediated ubiquitination therefore impairs hypoxia-induced selective mitophagy. In this study, we found that MUL1 overexpression mainly generated K29- and K48-linked ubiquitin chains which were also found in DRP1 ubiquitination chain (data not shown in the manuscript).

Figure 1. DRP1 ubiquitin chain linkages by MUL1. Cells were transfected with the expression vectors of DRP1, MUL1, and HA-ubiquitin WT vs variants that contain only one lysine as indicated. Cells were harvested by 2x Laemmli buffer and heat-inactivated. The diluted and cleared supernatants were immunoprecipitated by HA beads, followed by Western blot using antibody of DRP1 and MUL1.

In discussion (page 21), we added the statement that MUL1 overexpression gives DRP1 K48-linked ubiquitin chain (data not shown), which mediates ubiquitinated proteins for proteasomal degradation. Therefore, it is likely that DRP1 binding at the damage sites, a PA-enriched microenvironment, without NME3 protection is destabilized by MUL1 in NME3 deficient cells.

REVIEWERS' COMMENTS

Reviewer #1 (Remarks to the Author):

No further comments.